# Navigating Concept Drift and Temporal Shift: Distribution Shift Generalized Time-Series Forecasting

## Abstract

Time-series forecasting finds broad applications in real-world scenarios. Due to the dynamic nature of time series data, it is crucial for time-series forecasting models to produce robust predictions under potential distribution shifts. In this paper, we initially identify two types of distribution shifts in time series: concept drift and temporal shift. We acknowledge that while existing studies primarily focus on addressing temporal shift issues in time series, designing proper concept drift methods for time series data received comparatively less attention.

Motivated by the need to mitigate potential concept drift issues in time-series forecasting, this work proposes a novel soft attention mechanism that effectively leverages and ensemble information from the horizon time series. Furthermore, recognizing that both concept drift and temporal shift could occur concurrently in time-series forecasting scenarios while an integrated solution remains missing, this paper introduces `ShifTS`, a model-agnostic framework seamlessly addressing both concept drift and temporal shift issues in time-series forecasting. Extensive experiments demonstrate the efficacy of `ShifTS` in consistently enhancing the forecasting accuracy of agnostic models across multiple datasets, and consistently outperforming existing concept drift, temporal shift, and combined baselines.

## 1 Introduction

Time-series forecasting finds applications in various real-world scenarios such as economics, urban computing, and epidemiology (Zhu & Shasha, 2002; Zheng et al., 2014; Deb et al., 2017; Mathis et al., 2024). These applications involve predicting future trends or events based on historical time-series data. For example, economists use forecasts to make financial and marketing plans, while sociologists use them to allocate resources and formulate policies for traffic or disease control.

The recent advent of deep learning has revolutionized time-series forecasting, resulting in series of advanced forecasting models (Lai et al., 2018; Torres et al., 2021; Salinas et al., 2020; Nie et al., 2023; Zhou et al., 2021). However, despite these success, time-series forecasting faces certain challenges from distribution shifts due to the dynamic and complex nature of time series data. The distribution shifts in time series can be categorized into two types (Granger, 2003). First, the data distributions of the time series data themselves can change over time, including shifts in mean, variance, and autocorrelation structure, which is referred to as non-stationarity or temporal drift issues in time-series forecasting (Shimodaira, 2000; Du et al., 2021). For example, in influenza-like illness (ILI) forecasting, the distribution of influenza cases varies between summer and winter, with higher infection rates typically observed during the winter seasons. Second, time-series forecasting is compounded by unforeseen exogenous factors, which shifts the distribution of target time series. A prominent example is the COVID-19 pandemic, which led to an abnormal excess of influenza cases than normal years. These types of phenomena, categorized as concept drift problems in time-series forecasting (Gama et al., 2014; Lu et al., 2018), make it even more challenging.

While prior research has investigated strategies to mitigate temporal shifts (Liu et al., 2022; Kim et al., 2021; Fan et al., 2023), addressing concept drift issues in time-series forecasting has been largely overlooked. Although concept drift is a well-studied problem in general machine learning (Sagawa et al., 2019; Arjovsky et al., 2019; Ahuja et al., 2021), adapting these solutions to time-series

forecasting is challenging. Many of these methods require environment labels, which are typically unavailable in time-series datasets (Liu et al., 2024a). Indeed, the few concept drift approaches developed for time-series data are designed exclusively for online settings (Guo et al., 2021), limiting their generalizability to standard time-series forecasting tasks. Moreover, while both concept drift and temporal shift can simultaneously impact time-series forecasting, as shown in the previous ILI forecasting example, few existing researches or practical solutions address both issues together.

We aim to close this gap in the literature in this paper - this study aims to design an integrated framework that effectively addresses both concept drift, which has not been studied well by itself, and temporal shift. Our method involves ensembling time series across multiple horizon time steps to enhance generalization and mitigate concept drift, with seamless integration with normalization strategies to address temporal shift. The contributions of this paper are:

1. **Concept Drift for Time-Series:** We introduce soft attention masking (`SAM`) designed to mitigate concept drift issues by effectively using exogenous information from the horizon window. The soft attention allows the time-series forecasting models to weigh the ensemble of the time series at multiple horizon time steps to enhance the generalization ability.

2. **Integrated Framework:** We propose `ShifTS`, a practical and model-agnostic framework that tackles both concept drift and temporal shift in time-series forecasting tasks. `ShifTS` seamlessly integrates the proposed soft attention mechanism with established temporal shift mitigation techniques, facilitating enhanced forecasting accuracy.

3. **Comprehensive Evaluations:** We conduct extensive experiments on various time series datasets with multiple advanced time-series forecasting models. The proposed `ShifTS` demonstrates effectiveness by consistent performance improvements to agnostic forecasting models, as well as outperforming distribution shift baselines in better forecasting accuracy.

## 2 RELATED WORKS

**Time-Series Forecasting.** Classical statistical time-series forecasting models, such as ARIMA (Hyndman & Athanasopoulos, 2018), often face limitations in capturing complicated patterns and dependencies due to inherent model constraints (Nadaraya, 1964; Williams & Rasmussen, 1995; Smola & Schölkopf, 2004). Recent works in deep learning have achieved notable achievements in time-series forecasting, such as RNNs, LSTNet, N-BEATS (Sherstinsky, 2020; Lai et al., 2018; Oreshkin et al., 2020). State-of-the-art models build upon the successes of self-attention mechanisms (Vaswani et al., 2017) with transformer-based architectures and significantly improve forecasting accuracy, such as Informer, Autoformer, Fedformer, PatchTST, iTransformer, FRNet (Zhou et al., 2021; Wu et al., 2021; Zhou et al., 2022; Nie et al., 2023; Liu et al., 2024b; Zhang et al., 2024). However, these advanced models primarily rely on empirical risk minimization (ERM) with IID assumptions, i.e., train and test dataset follows the same data distribution, which exhibits limitations when potential distribution shifts in time series.

**Distribution Shift in Time-Series Forecasting.** In recent decades, learning under non-stationary distributions, where the target distribution over instances changes with time, has attracted attention within learning theory (Kuh et al., 1990; Bartlett, 1992). In the context of time series, the distribution shift can be categorized into concept drift and temporal shifts.

General concept drift methods (Arjovsky et al., 2019; Ahuja et al., 2021; Krueger et al., 2021; Pezeshki et al., 2021; Sagawa et al., 2019) assume instances sampled from various environments and propose to identify and utilize invariant predictors across these environments. However, when applied to time-series forecasting, these methods encounter limitations. Additional methods specifically tailored for time series data also encounter certain constraints: DIVERSITY (Lu et al., 2023) is designed for time series classification and detection only. OneNet (Wen et al., 2024) is tailored solely for online forecasting scenarios using online ensembling. PeTS (Zhao et al., 2023) focuses on distribution shifts induced by the specific phenomenon of performativity.

Other works specifically crafted for time-series forecasting aim to address temporal shift issues (Kim et al., 2021; Liu et al., 2022; Fan et al., 2023; Liu et al., 2023). These approaches implement carefully crafted normalization strategies to ensure that both the lookback and horizon of a univariate time series adhere to normalized distributions. This alignment helps alleviate potential temporal shifts, where the statistical properties of the lookback and horizon time series may differ, over time.

## 3 PROBLEM FORMULATION

### 3.1 TIME-SERIES FORECASTING

Time-series forecasting involves predicting future values of one or more dependent time series based on historical data, potentially augmented with exogenous covariate features. Let denote the target time series as $\mathbf{Y}$ and its associated exogenous covariate features as $\mathbf{X}$. At any time step $t$, time-series forecasting aims to predict $\mathbf{Y}_t^H = [yt + 1, y_{t+2}, \ldots, y_{t+H}] \in \mathbf{Y}$ using historical data $(\mathbf{X}_t^L, \mathbf{Y}_t^L)$, where $L$ represents the length of the historical data window, known as the *lookback window*, and $H$ denotes the forecasting time steps, known as the *horizon window*. Here, $\mathbf{X}_t^L = [x_{t-L+1}, x_{t-L+2}, \ldots, x_t] \in \mathbf{X}$ and $\mathbf{Y}_t^L = [y_{t-L+1}, y_{t-L+2}, \ldots, y_t] \in \mathbf{Y}$. For simplicity, we denote $\mathbf{Y}^H = \{\mathbf{Y}_t^H\}$ for $\forall t$ as the collection of horizon time-series of all time steps, and similar for $\mathbf{Y}^L$ and $\mathbf{X}^L$. Conventional approaches to time-series forecasting involve learning a model parameterized by $\theta$ through empirical risk minimization (ERM) to obtain $f_\theta : (\mathbf{X}^L, \mathbf{Y}^L) \to \mathbf{Y}^H$ for all time steps $t$.

In this study, we focus on univariate time-series forecasting with exogenous features, where $d_{\mathbf{Y}} = 1$ and $d_{\mathbf{X}} \geq 1$. Our methodology and this setup can be extended to multivariate time-series forecasting by employing multiple univariate forecastings (Lim & Zohren, 2021; Gruver et al., 2024).

### 3.2 DISTRIBUTION SHIFT IN TIME SERIES

Given the time-series forecasting setups, a time-series forecasting model aims to predict the target distribution $\mathrm{P}(\mathbf{Y}^H) = \mathrm{P}(\mathbf{Y}^H|\mathbf{Y}^L)\mathrm{P}(\mathbf{Y}^L) + \mathrm{P}(\mathbf{Y}^H|\mathbf{X}^L)\mathrm{P}(\mathbf{X}^L)$, which should be generalizable for both training and testing time steps. However, due to the dynamic nature of time-series data, forecasting faces challenges from distribution shifts, categorized into two types: temporal shift and concept drift. These two types of distribution shifts are defined as follows:

**Definition 3.1 (Temporal Shift (Shimodaira, 2000; Du et al., 2021))** *Temporal shift (also known as virtual shift (Tsymbal, 2004)) refers to the marginal probability distributions can change over time, and the conditional distributions are the same.*

**Definition 3.2 (Concept Drift (Lu et al., 2018))** *Concept drift (also known as real concept drift (Gama et al., 2014)) refers to the conditional distributions can change over time, and the marginal probability distributions are the same.*

Intuitively, a temporal shift indicates unstable marginal distributions (e.g. $\mathrm{P}(\mathbf{Y}^H) \neq \mathrm{P}(\mathbf{Y}^L)$), while a concept drift indicates unstable conditional distributions ($\mathrm{P}(\mathbf{Y}_i^H|\mathbf{X}_i^L) \neq \mathrm{P}(\mathbf{Y}_j^H|\mathbf{X}_j^L)$ for some $i, j \in t$). Existing methods for distribution shifts in time-series forecasting typically focus on mitigating temporal shifts through normalization, ensuring $\mathrm{P}(\mathbf{Y}^H) = \mathrm{P}(\mathbf{Y}^L)$ by both normalizing to standard 0-1 distributions (Kim et al., 2021; Liu et al., 2022; Fan et al., 2023).

Nevertheless, in addition to temporal shift, time-series forecasting also faces challenges from concept drift: The correlations between $\mathbf{X}$ and $\mathbf{Y}$ can change over time, making the conditional distributions $\mathrm{P}(\mathbf{Y}^H|\mathbf{X}^L)$ unstable and less predictable. Moreover, $\mathbf{X}^L$ may not fully explain or determine $\mathbf{Y}^H$, meaning that modeling the relationship solely through $\mathrm{P}(\mathbf{Y}^H|\mathbf{X}^L)$ may fail to capture the true correlations between $\mathbf{X}$ and $\mathbf{Y}$. A demonstration visualizing the differences and relationships between temporal shift and concept drift is provided in Appendix A.

While the concept drift issue has received considerable attention in existing studies on general machine learning, applying existing methods to time-series forecasting tasks presents certain challenges. Firstly, these methods typically rely on explicit environment labels as input (e.g., labeled rotation or noisy images in image classification), which are not readily available in time series datasets. Secondly, existing concept drift methods often require leveraging all correlated exogenous features to the target variable (Liu et al., 2024a), which may not be adequately captured in time series datasets (e.g., weather conditions affecting ILI forecasting, but not included in the current ILI dataset). Additionally, while both temporal shift and concept drift can manifest simultaneously in time-series forecasting (e.g., when both $\mathrm{P}(\mathbf{Y}^H) \neq \mathrm{P}(\mathbf{Y}^L)$ and $\mathrm{P}(\mathbf{Y}^H|\mathbf{X}^L)$ are unstable), few existing solutions effectively addresses both issues in the context of time-series forecasting.

# 4 METHODOLOGY

## 4.1 METHODOLOGY OVERVIEW

The high-level idea of our methodology lies in effectively harnessing information from the horizon window through soft attention masking SAM to mitigate concept drift in time-series forecasting. Moreover, acknowledging the absence of an integrated framework capable of addressing both temporal shift and concept drift within a single solution, we introduce a model-agnostic framework ShifTS tailored to tackle both challenges in time-series forecasting.

## 4.2 MITIGATING CONCEPT DRIFT

As defined in Definition 3.2, concept drift in time-series refers to the changing correlations between $\mathbf{X}$ and $\mathbf{Y}$ over time ($P(\mathbf{Y}_i^H|\mathbf{X}_i^L) \neq P(\mathbf{Y}_j^H|\mathbf{X}_j^L)$ for $i, j \in t$), which introduces instability when when modeling conditional distribution $P(\mathbf{Y}^H|\mathbf{X}^L)$. This instability in time-series forecasting arises from the insufficient information in $\mathbf{X}^L$ to fully determine $\mathbf{Y}^H$. Conventional concept drift methods necessarily assume that the inputs contain sufficient information to predict the output (Sagawa et al., 2019; Arjovsky et al., 2019), which may not always be valid in this context.

For example, an influenza-like illness (ILI) outbreak can be caused by multiple factors, including either extremely cold winter or hot summer weather (Nielsen et al., 2011; Jaakkola et al., 2014). In such cases, the stable conditional distribution to predict a winter ILI outbreak is $P(\mathbf{Y}^H = \text{outbreak}|\mathbf{X}^L = $ hot, or $\mathbf{X}^H = $ cold). However, without considering $\mathbf{X}^H$, modeling $P(\mathbf{Y}^H|\mathbf{X}^L)$ can become unstable, as $\mathbf{X}^L$ alone may not sufficiently determine $\mathbf{Y}^H$. That is, both $P(\mathbf{Y}^H = \text{outbreak}|\mathbf{X}^L = \text{hot})$ and $P(\mathbf{Y}^H = \text{outbreak}|\mathbf{X}^L \neq \text{hot})$ are possible, causing unstable conditional distributions over years.

To address unstable conditional distributions over time, we propose SAM, which mitigates concept drift by employing a weighted ensemble of multiple conditional distributions across the horizon. The intuition behind SAM is twofold: (1) Given that $\mathbf{X}^L$ alone cannot sufficiently determine $\mathbf{Y}^H$, SAM incorporates both lookback and horizon information from exogenous features to improve target prediction. This enables modeling multiple conditional distributions with inputs containing sufficient information to determine $\mathbf{Y}^H$, specifically $[P(\mathbf{Y}_t^H|\mathbf{X}_t^L), P(\mathbf{Y}_t^H|\mathbf{X}_{t+1}^L), \cdots, P(\mathbf{Y}_t^H|\mathbf{X}_{t+H}^L)]$ at each time step $t$. (2) Once sufficient determination is achieved through multiple conditional distributions, SAM uses soft attention masking to

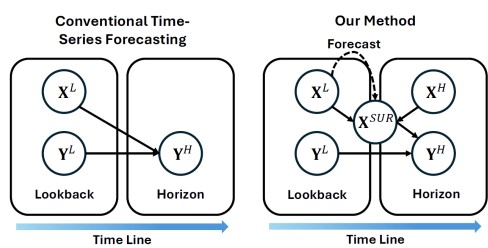

Figure 1: Comparison between conventional time-series forecasting and SAM. SAM aggregates both lookback and horizon information into $\mathbf{X}^{SUR}$ to identify stable aggregated conditional distributions and mitigate concept drift.

identify and aggregate those distributions that remain stable over time. Conditional distributions exhibiting variant patterns are learned with lower attention weights during empirical risk minimization and can be filtered via sparsity regularization, while those with high attention weights are recognized as invariant patterns, which remain unchanged during test time steps. Figure 1 illustrates the difference between SAM and conventional time-series forecasting from a causal perspective.

SAM operates through the following steps: First, it concatenates $[\mathbf{X}^L, \mathbf{X}^H]$ to form an entire time series of length $L + H$. Second, it slices the entire time series using a sliding window of size $H$, resulting in $L + 1$ slices (candidates). Next, it applies a learnable soft attention mask $\mathcal{M}$ to weigh and ensemble all slices, producing the ensembled time series $\mathbf{X}^{\text{SUR}}$, which is the surrogate exogenous time series that sufficiently supports and predicts the target series $\mathbf{Y}^H$. We denote this process as SAM $([\mathbf{X}^L, \mathbf{X}^H])$, and can be mathematically described as:

$$\mathbf{X}^{\text{SUR}} = \text{SAM}([\mathbf{X}^L, \mathbf{X}^H]) = \sum_{L+1} \mathcal{M}(\text{Slice}([\mathbf{X}^L, \mathbf{X}^H])) \tag{1}$$

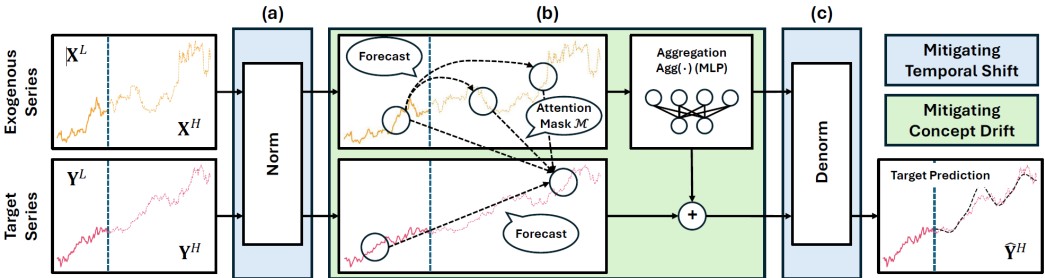

Figure 2: Diagram of `ShifTS`, consisting of three components: (a) normalization at the start (c) denormalization at the end to address temporal shifts, and (b) a two-stage forecasting process-The first stage predicts surrogate exogenous features, $\hat{\mathbf{X}}^{\text{SUR}}$, identified by the SAM, which capture invariant patterns essential for forecasting the target; The second stage uses both the predicted surrogate exogenous features and the original $\mathbf{Y}^L$ to predict $\mathbf{Y}^H$.

where $\text{Slice}(\cdot)$ denotes the sliding window process (i.e., slicing the time series $[L + H, d_{\mathbf{X}}] \rightarrow [H, L + 1, d_{\mathbf{X}}]$), and $\mathcal{M} \in \mathbb{R}^{L+1 \times d_{\mathbf{X}}}$ is the learnable soft attention mask with sparsity regularization:

$$\begin{aligned} \text{Softmax}: \quad & \mathcal{M}_j = \text{Softmax}(\mathcal{M}_j) \\ \text{Sparsity}: \quad & \mathcal{M}_{ij} = \mathcal{M}_{ij} \cdot \mathbb{1}_{(\mathcal{M}_{ij} - \mu(\mathcal{M}_j)) \geq 0} \\ \text{Normalize}: \quad & \mathcal{M}_j = \frac{\mathcal{M}_j}{|\mathcal{M}_j|} \end{aligned} \tag{2}$$

where $i, j$ are the first and second dimensions of $\mathcal{M}$. The intuition behind sparsity regularization is to filter out variant conditional distributions with learned attention weights, leaving only invariant ones, which are to be unchanged during testing. In practice, $\mathbf{X}^{\text{SUR}}$ may include horizon information that is unavailable during testing. Therefore, SAM estimates the surrogate features $\hat{\mathbf{X}}^{\text{SUR}}$ with agnostic forecasting models. The surrogate loss that aims to estimate $\hat{\mathbf{X}}^{\text{SUR}}$ is defined as:

$$\mathcal{L}_{\text{SUR}} = \text{MSE}(\mathbf{X}^{\text{SUR}}, \hat{\mathbf{X}}^{\text{SUR}}) \tag{3}$$

## 4.3 MITIGATING TEMPORAL SHIFT

Mitigating temporal shifts (referred to as 'non-stationary' problems in related literature) has received significant attention in the time-series forecasting community. The core intuition behind popular methods for addressing temporal shifts is to normalize data distributions before processing by the model and to denormalize the outputs afterward. This approach allows the normalized sequences to maintain more consistent mean and variance between the inputs and outputs of the forecasting model, i.e., $P(\mathbf{X}_{\text{Norm}}^L) \approx P(\mathbf{X}_{\text{Norm}}^H) \sim \text{Dist}(0, 1)$ and $P(\mathbf{Y}_{\text{Norm}}^L) \approx P(\mathbf{Y}_{\text{Norm}}^H) \sim \text{Dist}(0, 1)$, thus mitigating temporal shifts (i.e., marginal distribution shifts over time).

While the primary contribution of this work focuses on mitigating concept drift in time-series forecasting, addressing temporal shift is also crucial for effectively mitigating concept drift. The underlying intuition is that SAM aims to learn invariant patterns that yield a stable conditional distribution $P(\mathbf{Y}^H|\mathbf{X}^{\text{SUR}})$. However, achieving this stability becomes challenging without fixing a stable marginal distribution (e.g., $P(\mathbf{Y}^H)$ or $P(\mathbf{X}^{\text{SUR}})$), as these marginal distributions may vary over time. Therefore, a natural solution is to learn the conditional distribution under standardized marginal distributions which is achieved by temporal shift methods through instance normalization techniques.

Among the various approaches, Reversible Instance Normalization (RevIN) (Kim et al., 2021) is particularly notable and is utilized in this work due to its simplicity and effectiveness. Advanced techniques, such as SAN Liu et al. (2023) and N-S Transformer Liu et al. (2022), also show promise in mitigating temporal shift but require modifications to forecasting models or pre-training strategies. Exploring these advanced temporal shift methods remains promising but is beyond the scope of this study.

### 4.4 ShifTS: The Integrated Framework

By integrating SAM to mitigate concept drift and RevIN to address temporal shift, we propose ShifTS, a comprehensive framework that addresses both challenges in time-series forecasting. ShifTS is also model-agnostic, as it processes to identify stable conditional distributions, which can be learned by any time-series forecasting model. The workflow of ShifTS is illustrated in Figure 2 and consists of the following steps: (1) Normalize the input time series; (2) Forecast exogenous features $\hat{\mathbf{X}}^{\text{SUR}}$ that sufficiently support the target series, as determined by SAM; (3) An aggregation MLP that uses $\hat{\mathbf{X}}^{\text{SUR}}$ to forecast the target, denoted as Agg$(\cdot)$ in Figure 2 and Algorithm 1; (4) Denormalize the output time series. Conceptually, steps 1 and 4 mitigate the temporal shift, step 2 addresses concept drift, and step 3 performs weighted aggregation of exogenous features to support the target series. The optimization objective of ShifTS is described as follows:

$$\mathcal{L} = \mathcal{L}_{\text{SUR}}(\mathbf{X}^{\text{SUR}}, \hat{\mathbf{X}}^{\text{SUR}}) + \mathcal{L}_{\text{TS}}(\mathbf{Y}^H, \hat{\mathbf{Y}}^H) \tag{4}$$

Here, $\mathcal{L}_{\text{SUR}}$ is the surrogate loss that encourages learning to forecast exogenous features that sufficiently support the target series, and $\mathcal{L}$TS is the MSE loss used in conventional time-series forecasting. The pseudo-code for training and testing ShifTS is provided in Algorithm 1.

---

**Algorithm 1** ShifTS

---

1: **Training: Require:** Training data $\mathbf{X}^L$, $\mathbf{X}^H$, $\mathbf{Y}^L$, $\mathbf{Y}^H$; Initial parameters $f_0$, $\mathcal{M}_0$, Agg$_0$; **Output:** Model parameter $f$, $\mathcal{M}$, Agg

2: For $i$ in range $(E)$:

3:    Normalization:              $[\mathbf{X}_{\text{Norm}}^L, \mathbf{Y}_{\text{Norm}}^L] = \text{Norm}([\mathbf{X}^L, \mathbf{Y}^L])$

4:    Time-series forecasting:     $[\hat{\mathbf{X}}_{\text{Norm}}^{\text{SUR}}, \hat{\mathbf{Y}}_{\text{Norm}}^H] = f_i([\mathbf{X}_{\text{Norm}}^L, \mathbf{Y}_{\text{Norm}}^L])$

5:    Exogenous feature aggregation:  $\hat{\mathbf{Y}}_{\text{Norm}}^H = \hat{\mathbf{Y}}_{\text{Norm}}^H + \text{Agg}_i(\hat{\mathbf{X}}_{\text{Norm}}^{\text{SUR}})$

6:    Denormalization:            $[\hat{\mathbf{X}}^{\text{SUR}}, \hat{\mathbf{Y}}^H] = \text{Denorm}([\hat{\mathbf{X}}_{\text{Norm}}^{\text{SUR}}, \hat{\mathbf{Y}}_{\text{Norm}}^H])$

7:    Obtain sufficient ex-features:  $\mathbf{X}^{\text{SUR}} = \text{SAM}([\mathbf{X}^L, \mathbf{X}^H])$

8:    Compute loss:               $\mathcal{L} = \mathcal{L}_{\text{SUR}}(\mathbf{X}^{\text{SUR}}, \hat{\mathbf{X}}^{\text{SUR}}) + \mathcal{L}_{\text{TS}}(\mathbf{Y}^H, \hat{\mathbf{Y}}^H)$

9:    Update model parameter:      $f_{i+1} \leftarrow f_i, \mathcal{M}_{i+1} \leftarrow \mathcal{M}_i, \text{Agg}_{i+1} \leftarrow \text{Agg}_i$

10: Final model parameters: $f \leftarrow f_E, \mathcal{M} \leftarrow \mathcal{M}_E, \text{Agg} \leftarrow \text{Agg}_E$

11: **Testing: Require:** Test data $\mathbf{X}^L$, $\mathbf{Y}^L$, **Output:** Forecast target $\hat{\mathbf{Y}}^H$

12:    Normalization:             $[\mathbf{X}_{\text{Norm}}^L, \mathbf{Y}_{\text{Norm}}^L] = \text{Norm}([\mathbf{X}^L, \mathbf{Y}^L])$

13:    Time-series forecasting:    $[\hat{\mathbf{X}}_{\text{Norm}}^{\text{SUR}}, \hat{\mathbf{Y}}_{\text{Norm}}^H] = f([\mathbf{X}_{\text{Norm}}^L, \mathbf{Y}_{\text{Norm}}^L])$

14:    Exogenous feature aggregation:  $\hat{\mathbf{Y}}_{\text{Norm}}^H = \hat{\mathbf{Y}}_{\text{Norm}}^H + \text{Agg}(\hat{\mathbf{X}}_{\text{Norm}}^{\text{SUR}})$

15:    Denormalization:           $[\hat{\mathbf{X}}^{\text{SUR}}, \hat{\mathbf{Y}}^H] = \text{Denorm}([\hat{\mathbf{X}}_{\text{Norm}}^{\text{SUR}}, \hat{\mathbf{Y}}_{\text{Norm}}^H])$

---

## 5 Experiments

### 5.1 Setup

**Datasets.** We conduct experiments using six time-series datasets as leveraged in Liu et al. (2024a): The daily reported currency exchange rates (**Exchange**) (Lai et al., 2018); The weekly reported influenza-like illness patients (**ILI**) (Kamarthi et al., 2021); Two-hourly/minutely reported electricity transformer temperature (**ETTh1/ETTh2** and **ETTm1/ETTm2**, respectively) (Zhou et al., 2021). We follow the established experimental setups and target variable selections in previous works(Wu et al., 2021; 2022; Nie et al., 2023; Liu et al., 2024b). Datasets such as Traffic (PeMS) (Zhao et al., 2017) and Weather (Wu et al., 2021) are excluded from our evaluations, as their time series exhibit near-stationary behavior, with only moderate distribution shift issues. Further details on the dataset differences are discussed in Appendix B.1.

**Baselines.** We include two types of baselines for comprehensive evaluation on ShifTS:

| Model | Crossformer (ICLR'23) | | | | PatchTST (ICLR'23) | | | | iTransformer (ICLR'24) | | | |
|---|---|---|---|---|---|---|---|---|---|---|---|---|
| Method | ERM | | ShifTS | | ERM | | ShifTS | | ERM | | ShifTS | |
| Dataset | MSE | MAE | MSE | MAE | MSE | MAE | MSE | MAE | MSE | MAE | MSE | MAE |
| ILI 24 | 3.409 | 1.604 | **0.674** | **0.590** | 0.772 | 0.634 | **0.656** | **0.618** | 0.824 | 0.653 | **0.799** | **0.642** |
| ILI 36 | 4.001 | 1.772 | **0.687** | **0.617** | 0.763 | 0.649 | **0.694** | **0.602** | 0.917 | 0.738 | **0.690** | **0.640** |
| ILI 48 | 3.720 | 1.724 | **0.652** | **0.611** | 0.753 | 0.692 | **0.654** | **0.630** | 0.772 | 0.699 | **0.680** | **0.665** |
| ILI 60 | 3.689 | 1.715 | **0.658** | **0.633** | 0.761 | 0.724 | **0.680** | **0.656** | 0.729 | 0.710 | **0.672** | **0.667** |
| ILI IMP. | | | 81.9% | 64.0% | | | 12.0% | 7.1% | | | 13.8% | 6.5% |
| Exchange 96 | 0.338 | 0.475 | **0.102** | **0.237** | 0.130 | 0.265 | **0.102** | **0.236** | 0.135 | 0.272 | **0.115** | **0.255** |
| Exchange 192 | 0.566 | 0.622 | **0.203** | **0.338** | 0.247 | 0.394 | **0.194** | **0.332** | 0.250 | 0.376 | **0.209** | **0.343** |
| Exchange 336 | 1.078 | 0.867 | **0.407** | **0.484** | 0.522 | 0.557 | **0.388** | **0.477** | 0.450 | 0.503 | **0.426** | **0.495** |
| Exchange 720 | 1.292 | 0.963 | **1.165** | **0.813** | 1.171 | 0.824 | **0.995** | **0.747** | 1.501 | 0.941 | **1.138** | **0.827** |
| Exchange IMP. | | | 53.5% | 38.9% | | | 20.9% | 12.6% | | | 15.2% | 6.9% |
| ETTh1 96 | 0.145 | 0.312 | **0.055** | **0.180** | 0.064 | 0.193 | **0.056** | **0.181** | 0.061 | 0.190 | **0.056** | **0.181** |
| ETTh1 192 | 0.240 | 0.420 | **0.072** | **0.206** | 0.085 | 0.222 | **0.073** | **0.209** | 0.076 | 0.219 | **0.072** | **0.205** |
| ETTh1 336 | 0.240 | 0.424 | **0.084** | **0.228** | 0.096 | 0.244 | **0.089** | **0.235** | 0.086 | 0.227 | **0.083** | **0.225** |
| ETTh1 720 | 0.391 | 0.553 | **0.095** | **0.244** | 0.128 | 0.282 | **0.097** | **0.245** | 0.085 | 0.232 | **0.082** | **0.230** |
| ETTh1 IMP. | | | 68.2% | 48.8% | | | 14.5% | 7.2% | | | 5.1% | 3.3% |
| ETTh2 96 | 0.255 | 0.408 | **0.137** | **0.286** | 0.154 | 0.309 | **0.139** | **0.287** | 0.141 | 0.292 | **0.137** | **0.288** |
| ETTh2 192 | 1.257 | 1.034 | **0.182** | **0.338** | 0.204 | 0.374 | **0.191** | **0.345** | 0.194 | 0.347 | **0.184** | **0.339** |
| ETTh2 336 | 0.783 | 0.771 | **0.234** | **0.388** | 0.252 | 0.406 | **0.222** | **0.381** | 0.229 | 0.383 | **0.225** | **0.381** |
| ETTh2 720 | 1.455 | 1.100 | **0.234** | **0.389** | 0.259 | 0.411 | **0.236** | **0.390** | 0.266 | 0.413 | **0.235** | **0.390** |
| ETTh2 IMP. | | | 71.4% | 52.9% | | | 9.2% | 6.5% | | | 5.4% | 2.5% |
| ETTm1 96 | 0.050 | 0.174 | **0.028** | **0.126** | 0.031 | 0.135 | **0.029** | **0.128** | **0.030** | **0.131** | 0.030 | 0.131 |
| ETTm1 192 | 0.271 | 0.454 | **0.043** | **0.158** | 0.048 | 0.166 | **0.044** | **0.161** | 0.049 | 0.171 | **0.046** | **0.165** |
| ETTm1 336 | 0.731 | 0.805 | **0.057** | **0.184** | **0.058** | 0.190 | 0.058 | **0.186** | 0.066 | 0.199 | **0.059** | **0.188** |
| ETTm1 720 | 0.829 | 0.849 | **0.083** | **0.219** | 0.083 | 0.223 | **0.080** | **0.219** | 0.082 | 0.219 | **0.079** | **0.217** |
| ETTm1 IMP. | | | 77.3% | 61.0% | | | 4.6% | 3.0% | | | 5.1% | 2.5% |
| ETTm2 96 | 0.153 | 0.315 | **0.069** | **0.190** | 0.078 | 0.206 | **0.067** | **0.188** | **0.073** | 0.200 | 0.073 | **0.195** |
| ETTm2 192 | 0.408 | 0.526 | **0.105** | **0.242** | 0.113 | 0.246 | **0.101** | **0.237** | 0.119 | 0.251 | **0.108** | **0.248** |
| ETTm2 336 | 0.428 | 0.504 | **0.146** | **0.289** | 0.176 | 0.320 | **0.134** | **0.278** | 0.157 | 0.302 | **0.144** | **0.291** |
| ETTm2 720 | 1.965 | 1.205 | **0.191** | **0.342** | 0.220 | 0.368 | **0.185** | **0.334** | 0.196 | 0.347 | **0.193** | **0.344** |
| ETTm2 IMP. | | | 71.3% | 52.0% | | | 15.9% | 8.6% | | | 4.8% | 2.1% |

Table 1: Performance comparison on forecasting errors without (ERM) and with `ShifTS`. Employing `ShifTS` shows consistent performance gains agnostic to forecasting models. The top-performing method is in bold. 'IMP.' denotes the average improvements over all horizons of `ShifTS` vs ERM.

**Forecasting Model Baselines**: `ShifTS` is model-agnostic, we include six time-series forecasting models (referred to as 'Model' in Table 1 and 4), including: **Informer** (Zhou et al., 2021), **Pyraformer** (Liu et al., 2021), **Crossformer** (Zhang & Yan, 2022), **PatchTST** (Nie et al., 2023), **TimeMixer** (Wang et al., 2024) and **iTransformer** (Liu et al., 2024b), which of the last two are the state-of-the-art (SOTA) forecasting model. These models are used to demonstrate that `ShifTS` consistently enhances forecasting accuracy across various models, including SOTA.

**Distribution Shift Baselines**: We compare `ShifTS` with various distribution shift methods (referred to as 'Method' in Table 2): (1) Three non-stationary methods for addressing temporal distribution shifts in time-series forecasting **N-S Trans.** (Liu et al., 2022), **RevIN** (Kim et al., 2021), and **SAN** (Liu et al., 2023). We omit **Dish-TS** (Fan et al., 2023) and **SIN** (Han et al., 2024) from the main text due to their instability on univariate targets. (2) Four concept drift methods, including **GroupDRO** (Sagawa et al., 2019), **IRM** (Arjovsky et al., 2019), **VREx** (Krueger et al., 2021), and **EIIL** (Creager et al., 2021), which are primarily designed for general applications. (3) Three combined methods for both temporal distribution shifts and concept drift: **IRM+RevIN**, **EIIL+RevIN**, and SOTA time-series distribution shift method **FOIL** (Liu et al., 2024a). These comparisons aim to highlight the advantages of `ShifTS` in distribution shift generalization over existing distribution shift approaches.

**Evaluation.** We measure the forecasting errors using mean squared error (**MSE**) and mean absolute error (**MAE**). The formula of the metrics are: MSE $= \frac{1}{n} \sum_{i=1}^{n} (\boldsymbol{y} - \hat{\boldsymbol{y}})^2$ and MSE $= \frac{1}{n} \sum_{i=1}^{n} |\boldsymbol{y} - \hat{\boldsymbol{y}}|$. The proposed `ShifTS` does not introduce any additional hyperparameter beyond those inherent in the forecasting models. Therefore, we omit the hyperparameter sensitivity study in our experiments.

**Reproducibility.** All models are trained on NVIDIA Tesla V100 32GB GPUs. All training data and code are anonymously available at: `https://anonymous.4open.science/r/shifts_iclr-56A0`. More experiment details are presented in Appendix B.2.

| Dataset | | ILI | | Exchange | | ETTh1 | | ETTh2 | |
|---|---|---|---|---|---|---|---|---|---|
| Method | | MSE | MAE | MSE | MAE | MSE | MAE | MSE | MAE |
| Base | ERM | 3.705 | 1.704 | 0.819 | 0.732 | 0.254 | 0.427 | 0.937 | 0.828 |
| Concept Drift Method | GroupDRO | 2.285 | 1.287 | 0.821 | 0.751 | 0.278 | 0.453 | 1.150 | 0.936 |
| | IRM | 2.248 | 1.237 | 0.846 | 0.754 | 0.201 | 0.367 | 0.878 | 0.792 |
| | VREx | 2.285 | 1.286 | 0.821 | 0.742 | 0.314 | 0.486 | 1.142 | 0.938 |
| | EIIL | 2.036 | 1.159 | 0.822 | 0.749 | 0.212 | 0.433 | 1.122 | 0.930 |
| Temporal Shift Method | RevIN | 0.815 | 0.708 | 0.475 | 0.476 | 0.085 | 0.224 | 0.205 | 0.358 |
| | N-S Trans. | 0.781 | 0.688 | 0.484 | 0.481 | 0.086 | 0.226 | 0.203 | 0.355 |
| | SAN | 0.757 | 0.715 | **0.415** | **0.453** | 0.088 | 0.225 | 0.199 | 0.348 |
| Combined Method | IRM+RevIN | 0.809 | 0.711 | 0.481 | 0.476 | 0.089 | 0.231 | 0.202 | 0.362 |
| | EIIL+RevIN | 0.799 | 0.706 | 0.483 | 0.485 | 0.085 | 0.225 | 0.218 | 0.380 |
| | FOIL | 0.735 | 0.651 | 0.497 | 0.481 | 0.081 | 0.219 | 0.206 | 0.357 |
| | `ShifTS` (Ours) | **0.668** | **0.613** | 0.470 | 0.468 | **0.076** | **0.214** | **0.194** | **0.348** |

Table 2: Averaged performance comparison between `ShifTS` and distribution shift baselines with Crossformer. `ShifTS` achieves the best and second-best performance in 6 and 2 out of 8 evaluations. The best results are highlighted in bold and the second-best results are underlined.

## 5.2 PERFORMANCE IMPROVEMENT ACROSS BASE FORECASTING MODELS

To showcase the effectiveness of `ShifTS` in reducing forecasting errors, we conduct experiments to compare performance with and without the inclusion of `ShifTS` across various time series datasets and forecasting horizons, utilizing five transformer-based forecasting models. Evaluation results for Crossformer, PatchTST, and iTransformer are presented in Table 1. Additional evaluations for older models, including Informer, Pyraformer, and TimeMixer, are provided in Table 4 in Appendix C.1.

The results highlight the effectiveness of `ShifTS` in consistent performance improvements over agnostic forecasting models. articularly remarkable is its ability to consistently enhance performance, even when incorporated with advanced models like iTransformer, yielding reductions of up to 15% in forecasting errors. Moreover, `ShifTS` demonstrates heightened effectiveness when applied to other non-state-of-the-art forecasting models, such as Informer and PatchTST.

In addition to the observed performance improvements, our results reveal two further insights:

**The effectiveness of `ShifTS` relies on the insights provided by the horizon data.** The performance enhancements exhibit variations across different datasets. For instance, the application of `ShifTS` on ILI and Exchange datasets yields greater performance improvements compared to ETT datasets overall. To interpret the phenomenon and determine the conditions under which `ShifTS` could be most effective in practical scenarios, we quantify the mutual information $I(\mathbf{X}^H; \mathbf{Y}^H)$ shared between $\mathbf{X}^H$ and $\mathbf{Y}^H$ (detailed setup provided in Appendix B.2). We plot the relationship between $I(\mathbf{X}^H; \mathbf{Y}^H)$ and performance gains in Figure 3(a). The scatter plot illustrates a positive linear correlation between $I(\mathbf{X}^H; \mathbf{Y}^H)$ and performance gains, supported by a p-value $p = 0.012 \leq 0.05$. This observation suggests that the greater the amount of useful information from exogenous features within the horizon window, the more substantial the performance gains achieved by `ShifTS`. This insight aligns with the innovation of `ShifTS`, which is to comprehensively exploit and leverage information from the horizon window, which has been overlooked by existing methodologies.

**The extent of quantitative performance gains achieved by `ShifTS` depends on the underlying forecasting model.** Notably, the extent of performance enhancements achieved by `ShifTS` varies across different forecasting models. For example, the performance gains on the simpler Informer model by `ShifTS` is more significant than the SOTA iTransformer model. Importantly, we emphasize two key observations: Firstly, even when applied to the iTransformer model, `ShifTS` demonstrates a notable performance boost of approximately 15% on both ILI and Exchange datasets, consistent with the aforehead intuition. Secondly, integrating `ShifTS` into forecasting processes should, at the very least, maintain or improve the performance of standalone forecasting models, as evidenced by consistent performance enhancements observed across all datasets with iTransformer model.

## 5.3 COMPARISON WITH DISTRIBUTION SHIFT METHODS

To illustrate the advantages of `ShifTS` over other model-agnostic methods for addressing distribution shifts, we conduct experiments to compare performance across distribution shift baselines

following Liu et al. (2024a), where the evaluations on minutely ETT datasets were omitted, as their data characteristics and forecasting quality generally align with those of hourly ETT datasets. We use Crossformer as the forecasting model. The averaged results are summarized in Table 2.

The results highlight the advantages of `ShifTS` over existing distribution shift methods, achieving the highest average forecasting accuracy in 6 out of 8 evaluations, with the remaining 2 evaluations ranking second. Notably, as discussed in Section 4.3, `ShifTS` is flexible in integrating other advanced temporal shift methods to enhance performance. For instance, in the Exchange dataset, where SAN outperforms `ShifTS`, `ShifTS` can further improve its accuracy by incorpo-

| Horizon | ShifTS | SAN | ShifTS+SAN |
|---------|--------|-----|------------|
| 96 | 0.102 | 0.091 | **0.089** |
| 192 | 0.207 | 0.195 | **0.187** |
| 336 | 0.407 | 0.373 | **0.372** |
| 720 | 1.165 | 1.001 | **0.981** |
| Avg. | 0.470 | 0.415 | **0.407** |

Table 3: MSE comparison between `ShifTS`, SAN, and `ShifTS`+SAN on Exchange dataset. `ShifTS`+SAN achieves the best performance on all evaluations.

rating SAN in place of RevIN. Detailed MSE values are provided in Table 3. Additionally, the results illustrate the further benefits of addressing concept drift using `SAM` when temporal shift is effectively managed.

## 5.4 ABLATION STUDY

To demonstrate the effectiveness of each module in `ShifTS`, we conducted an ablation study using two modified versions: `ShifTS\TS` and `ShifTS\CD`. `ShifTS\TS` excludes the temporal shift adjustment via RevIN, while `ShifTS\CD` excludes the concept drift handling via `SAM`. Additionally, conventional forecasting models that do not address either concept drift or temporal shift are denoted as 'Base'. We performed experiments on the Exchange datasets using previous three baseline forecasting models, with a fixed forecasting horizon of 96. The results are visualized in Figure 3(b). The visualization reveals the following observations:

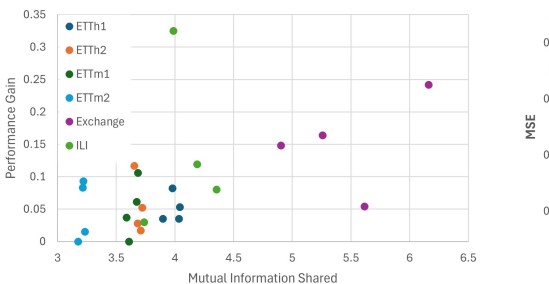 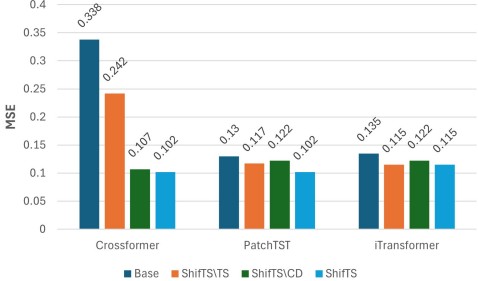

Figure 3: **Left (a):** The performance gains of `ShifTS` versus the mutual information shared between $\mathbf{X}^H$ and $\mathbf{Y}^H$. Greater mutual information in $\mathbf{X}^H$ compared to $\mathbf{Y}^H$ correlates with more significant performance gains achieved by `ShifTS`. **Right (b): Ablation Study.** Addressing either concept drift or temporal shift individually provides certain benefits in reducing forecasting error, but `ShifTS`, which tackles both, achieves the lowest forecasting error.

First, addressing both temporal shift and concept drift together, as implemented in `ShifTS`, yields lower forecasting errors than addressing only one type of distribution shift (`ShifTS\TS` and `ShifTS\CD`) or not considering any distribution shift adjustments (Base). This suggests that temporal shift and concept drift are likely interrelated and co-existed in time series data, and addressing both provides significant benefits.

Second, for forecasting models that inherently address temporal shift, such as PatchTST and iTransformer that incorporate norm/denorm, the performance gains from mitigating concept drift are more significant than those from additionally mitigating temporal shift using RevIN. In contrast, for models without any temporal shift mitigation, such as Crossformer, tackling temporal shift leads to a greater performance improvement than addressing concept drift. This distinction highlights the coexistence of both concept drift and temporal shift in time-series forecasting tasks. While handling temporal shifts is a fundamental challenge that has already received considerable attention, once resolved, mitigating

concept drift—an issue largely overlooked in current research and a unique key contribution of our work—can lead to promising improvements in forecasting accuracy.

## 6 CONCLUSION

In this paper, we identify that both concept drift and temporal shift issues can coexist in time series forecasting. While mitigating temporal shifts has received significant attention from the time-series forecasting community, concept drift issues have been largely neglected. To address this gap, we first propose a soft attention mechanism, SAM, which effectively mitigates concept drift in time-series forecasting by incorporating horizon information of exogenous features to enhance generalization ability. We then introduce ShifTS, a model-agnostic framework that tackles both concept drift and temporal shift issues. Our comprehensive evaluations demonstrate the effectiveness of ShifTS, and the benefit of SAM is further illustrated through an ablation study.

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

## A  Appendix A: Temporal Shift and Concept Drift

To highlight the differences between concept drift and temporal shift, we provide visualizations of both phenomena. Figure 4 illustrates temporal shift, while Figure 5 demonstrates concept drift[1].

Temporal shift refers to changes in the statistical properties of a univariate time series data, such as mean, variance, and autocorrelation structures, over time. For instance, the mean and variance of the given time series shift between lookback window and horizon window, as depicted in Figure 4. This issue is inherent in time series forecasting and can occur on any given time series data, regardless of whether the data pertains to the target series or exogenous features.

In contrast, concept drift describes to changes in the correlations between exogenous features and the target series over time. Figure 5 illustrates this phenomenon, where increases in exogenous features at earlier time steps lead to increases in the target series, while increases at later time steps result in decreases. Unlike temporal shift, concept drift involves multiple correlated time series and is not an inherent issue in univariate time series analysis.

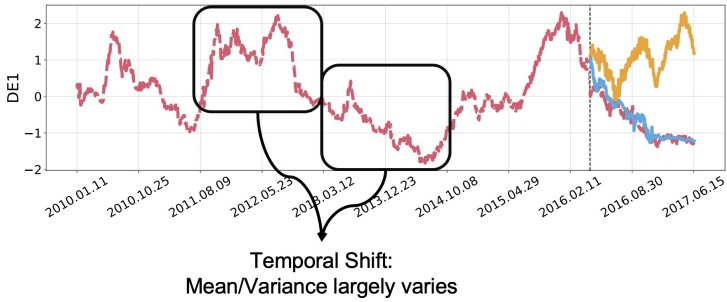

Figure 4: Demonstration of temporal shift phenomenon within time series data, showcasing the variations in statistical properties, including mean and variance, over time as the emergence of temporal shift (**Red:** ground truth; **Yellow:** N-BEATS prediction; **Blue:** N-BEATS+RevIN prediction).

---

[1]Figures adapted from: `https://github.com/ts-kim/RevIN`

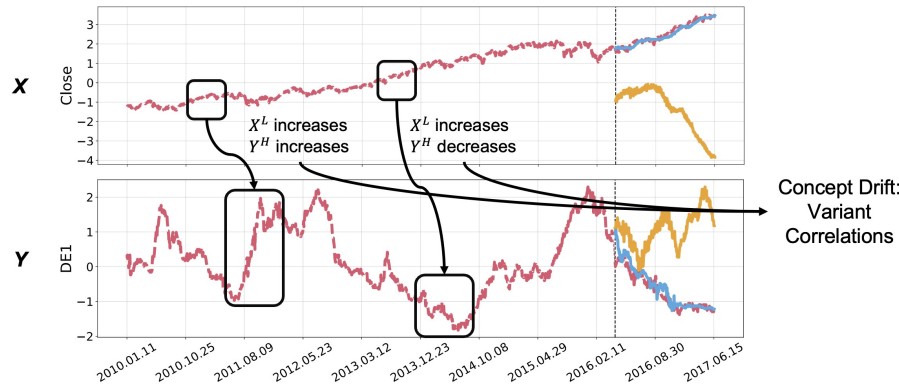

Figure 5: Demonstration of concept drift phenomenon within time series data, showcasing the variations in correlation structures between arget series **Y** and exogenous feature **X** over time as the emergence of concept drift (**Red:** ground truth; **Yellow:** N-BEATS prediction; **Blue:** N-BEATS+RevIN prediction).

## B    APPENDIX B: ADDITIONAL EXPERIMENT DETAILS

### B.1    DATASETS

We conduct experiments on six real-world datasets, which are commonly used as benchmark datasets:

- **ILI.** The ILI dataset collects data on influenza-like illness patients weekly, with eight variables.

- **Exchange.** The Exchange dataset records the daily exchange rate of eight currencies.

- **ETT.** The ETT dataset contains four sub-datasets: **ETTh1**, **ETTh2**, **ETTm1**, **ETTm2**. The datasets record electricity transformer temperatures from two separate counties in China (distinguished by '1' and '2'), with two granularities: minutely and hourly (distinguished by 'm' and 'h'). All sub-datasets have seven variables/features.

We follow Wu et al. (2022); Nie et al. (2023); Liu et al. (2024b) to preprocess data, which guides splitting datasets into train/validation/test sets and selecting the target variables. All datasets are preprocessed using the zero-mean normalization method.

Additional popular time-series datasets, such as Traffic (which records road occupancy rates from various sensors on San Francisco freeways), Electricity (which tracks hourly electricity consumption for 321 customers), and Weather (which collects 21 meteorological indicators in Germany, such as humidity and air temperature), are omitted from our evaluations. These datasets exhibit strong periodic signals and display near-stationary properties, making distribution shift issues less prevalent. A visualization comparison between the ETTh1 and Traffic datasets, shown in Figure 6, further supports this observation.

### B.2    BASELINE IMPLEMENTATION

We follow the commonly adopted setup for defining the forecasting horizon window length, as outlined in prior works Wu et al. (2022); Nie et al. (2023); Liu et al. (2024b). Specifically, for datasets such as ETT and Exchange, the forecasting horizon windows are chosen from the set [96, 192, 336, 720], with a fixed lookback window size of 96 and a consistent label window size of 48 for the decoder (if required). Similarly, for the weekly reported ILI dataset, we employ forecasting horizon windows from [24, 36, 48, 60], with a fixed lookback window size of 36 and a constant label window size of 18 for the decoder (if required).

In the context of concept drift baselines, several baselines like GroupDRO, IRM, and VREx require environment labels, which are typically absent in time series datasets. To address this, we partition the training set into $k$ equal-length time segments to serve as predefined environment labels.

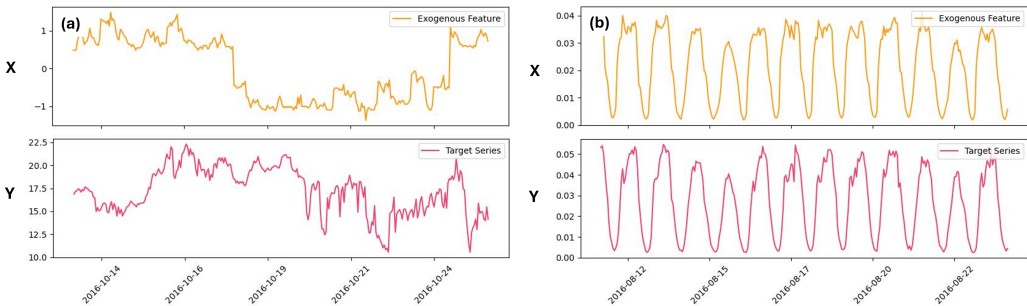

Figure 6: Distribution shift issues across datasets: **Left (a): ETT.** Both temporal shift and concept drift are present. The target series shows varying statistics over time (e.g., lower variance in earlier periods and higher variance later), causing temporal shift. The correlation between $\mathbf{X}$ and $\mathbf{Y}$ is unclear and unstable, causing concept drift. **Right (b): Traffic.** Both temporal shift and concept drift are moderate. The target series exhibits near-periodicity, making the temporal shift moderate. Moreover, the correlation between $\mathbf{X}$ and $\mathbf{Y}$ remains stable (e.g., both increase or decrease simultaneously), making concept drift moderate.

For baseline time-series forecasting models, we follow implementations and suggested hyperparameters (with additional tuning) sourced from the Time Series Library[2]. For concept drift baselines, we utilize implementations and hyperparameter tuning strategies recommended by DomainBed[3]. For temporal shift baselines, we adopt implementations and hyperparameter configurations outlined in their respective papers. Additionally, we add an additional MLP layer to the end PatchTST to effectively utilize exogenous features, following Liu et al. (2024a).

In the ablation study, for the implementation of PatchTST and iTransformer, we follow the original approach by applying norm and denorm operations to the 'Base' model. To clarify our notation, `ShifTS\TS` refers to the model with standard norm/denorm operations and `SAM`, while `ShifTS\CD` denotes the version where the regular norm/denorm is replaced with RevIN.

### B.3 MUTUAL INFORMATION VISUALIZATION

For a given time series dataset, we compute the mutual information $\boldsymbol{I}(\mathbf{X}^H; \mathbf{Y}^H)$ for each training time step and each exogenous feature dimension individually, following:

$$\boldsymbol{I}(\mathbf{X}^H; \mathbf{Y}^H) = \sum_{x \in \mathbf{X}^H} \sum_{y \in \mathbf{Y}^H} P(x,y) \log \frac{P(x,y)}{P(x)P(y)} \tag{5}$$

We then average the mutual information across all time steps for each exogenous feature dimension and identify the maximum averaged mutual information over all feature dimensions. This process allows us to assess the information content of each feature dimension in relation to the target series.

We visualize the maximum averaged mutual information plotted against the corresponding performance gain in Figure 3(a). This visualization provides insights into how the information content of different feature dimensions relates to the performance improvement achieved in the forecasting model.

## C    APPENDIX C: ADDITIONAL RESULTS

### C.1    EVALUATIONS ON AGNOSTIC PERFORMANCE GAINS

To further demonstrate the benefit of `ShifTS` in improving the forecasting accuracy over agnostic forecasting models, we additionally evaluate the performance differences without and with `ShifTS` on Informer, Pyraformer, and TimeMixer. The detailed results are presented in Table 4. The additional

---

[2]https://github.com/thuml/Time-Series-Library
[3]https://github.com/facebookresearch/DomainBed

| Model | Informer (AAAI'21) | | | | Pyraformer (ICLR'21) | | | | TimeMixer (ICLR'24) | | | |
|---|---|---|---|---|---|---|---|---|---|---|---|---|
| Method | ERM | | ShifTS | | ERM | | ShifTS | | ERM | | ShifTS | |
| Dataset | MSE | MAE | MSE | MAE | MSE | MAE | MSE | MAE | MSE | MAE | MSE | MAE |
| ILI 24 | 5.032 | 1.935 | **1.030** | **0.812** | 4.692 | 1.898 | **0.979** | **0.749** | 0.853 | 0.733 | **0.789** | **0.702** |
| ILI 36 | 4.475 | 1.876 | **1.046** | **0.850** | 4.814 | 1.950 | **0.866** | **0.740** | 0.721 | 0.676 | **0.697** | **0.665** |
| ILI 48 | 4.506 | 1.879 | **0.918** | **0.818** | 4.109 | 1.801 | **0.789** | **0.732** | **0.737** | **0.692** | 0.741 | 0.711 |
| ILI 60 | 4.313 | 1.850 | **0.957** | **0.839** | 4.483 | 1.850 | **0.723** | **0.698** | 0.788 | 0.723 | **0.670** | **0.659** |
| ILI IMP. | | | 78.4% | 56.0% | | | 81.5% | 61.1% | | | 6.3% | 3.0% |
| Exchange 96 | 0.839 | 0.746 | **0.137** | **0.277** | 0.410 | 0.525 | **0.145** | **0.275** | 0.127 | 0.268 | **0.098** | **0.234** |
| Exchange 192 | 0.862 | 0.773 | **0.210** | **0.346** | 0.529 | 0.610 | **0.300** | **0.404** | 0.229 | 0.355 | **0.214** | **0.352** |
| Exchange 336 | 1.597 | 1.063 | **0.378** | **0.485** | 0.851 | 0.778 | **0.440** | **0.506** | 0.553 | 0.560 | **0.440** | **0.491** |
| Exchange 720 | 4.358 | 1.935 | **0.760** | **0.655** | 1.558 | 1.067 | 1.509 | 0.963 | 1.173 | 0.834 | **0.962** | **0.747** |
| Exchange IMP. | | | 79.5% | 59.7% | | | 39.8% | 31.5% | | | 16.9% | 9.1% |
| ETTh1 96 | 0.891 | 0.863 | **0.095** | **0.231** | 0.653 | 0.748 | **0.065** | **0.197** | **0.059** | **0.184** | **0.059** | 0.187 |
| ETTh1 192 | 1.027 | 0.958 | **0.096** | **0.237** | 0.853 | 0.828 | **0.075** | **0.210** | 0.099 | 0.247 | **0.077** | **0.211** |
| ETTh1 336 | 1.055 | 0.961 | **0.092** | **0.237** | 0.705 | 0.797 | **0.092** | **0.238** | 0.121 | 0.279 | **0.098** | **0.246** |
| ETTh1 720 | 1.077 | 0.969 | **0.100** | **0.252** | 0.562 | 0.695 | **0.126** | **0.279** | 0.139 | 0.299 | **0.099** | **0.252** |
| ETTh1 IMP. | | | 90.7% | 74.5% | | | 86.4% | 69.6% | | | 23.3% | 10.1% |
| ETTh2 96 | 3.195 | 1.651 | **0.232** | **0.381** | 1.598 | 1.127 | **0.156** | **0.307** | 0.152 | 0.303 | **0.146** | **0.299** |
| ETTh2 192 | 3.569 | 1.778 | **0.334** | **0.464** | 3.314 | 1.599 | **0.217** | **0.367** | 0.195 | 0.349 | **0.185** | **0.343** |
| ETTh2 336 | 2.556 | 1.468 | **0.400** | **0.512** | 2.571 | 1.489 | **0.245** | **0.398** | 0.238 | 0.392 | **0.230** | **0.381** |
| ETTh2 720 | 2.723 | 1.532 | **0.489** | **0.579** | 2.294 | 1.409 | **0.261** | **0.410** | 0.273 | 0.421 | **0.249** | **0.397** |
| ETTh2 IMP. | | | 82.0% | 69.5% | | | 90.6% | 73.5% | | | 5.3% | 2.9% |
| ETTm1 96 | 0.320 | 0.433 | **0.055** | **0.175** | 0.130 | 0.298 | **0.028** | **0.125** | 0.030 | 0.128 | **0.029** | **0.126** |
| ETTm1 192 | 0.459 | 0.582 | **0.079** | **0.211** | 0.240 | 0.4112 | **0.045** | **0.162** | **0.047** | 0.165 | **0.047** | **0.164** |
| ETTm1 336 | 0.457 | 0.556 | **0.104** | **0.243** | 0.359 | 0.512 | **0.062** | **0.192** | 0.063 | 0.191 | **0.060** | **0.189** |
| ETTm1 720 | 0.735 | 0.760 | **0.148** | **0.294** | 0.657 | 0.750 | **0.091** | **0.231** | 0.083 | 0.223 | **0.081** | **0.220** |
| ETTm1 IMP. | | | 80.7% | 60.3% | | | 82.2% | 62.6% | | | 2.3% | 1.1% |
| ETTm2 96 | 0.191 | 0.345 | **0.154** | **0.298** | 0.275 | 0.422 | **0.075** | **0.200** | 0.079 | 0.205 | **0.075** | **0.201** |
| ETTm2 192 | 0.458 | 0.556 | **0.243** | **0.378** | 0.484 | 0.552 | **0.107** | **0.248** | 0.121 | 0.259 | **0.111** | **0.250** |
| ETTm2 336 | 0.606 | 0.624 | **0.515** | **0.539** | 1.138 | 0.909 | **0.146** | **0.293** | 0.150 | 0.295 | **0.148** | **0.294** |
| ETTm2 720 | 1.175 | 0.879 | **0.564** | **0.592** | 2.920 | 1.537 | **0.196** | **0.347** | 0.246 | 0.387 | **0.198** | **0.346** |
| ETTm2 IMP. | | | 33.4% | 23.0% | | | 82.8% | 63.2% | | | 8.5% | 4.1% |

Table 4: Performance comparison on forecasting errors without (ERM) and with `ShifTS` on Informer, Pyraformer, and TimeMixer. Employing `ShifTS` again shows near-consistent performance gains agnostic to forecasting models. The top-performing method is in bold. 'IMP.' denotes the average improvements over all horizons of `ShifTS` vs ERM.

evaluations again show consistent performance improvements on these models. Moreover, compared to the results in Table 1, the performance gains on these older models are even more significant. This observation highlight the needs of mitigating both concept drift and temporal shift in time-series forecasting, as such problem are rarely considered in these models, but the later models (e.g., PatchTST and iTransformer are compounded with normalizaiton/denormalizaiton processes).

# D  LIMITATION DISCUSSION

This work introduces `SAM` to address concept drift and proposes an integrated framework, `ShifTS`, which combines `SAM` with temporal shift mitigation techniques to enhance the accuracy of time-series forecasting. Extensive empirical evaluations support the effectiveness of these methods. However, the limitations of this study lie in two aspects: First, the distribution shift methods in time-series forecasting, including `ShifTS`, lack a theoretical guarantee. For example, no analysis quantifies how much the error bound can be tightened by addressing concept drift or temporal shift compared to vanilla time-series forecasting methods. Second, while this paper defines concept drift and temporal shift issues within the context of time-series forecasting, `SAM` and `ShifTS` are not the only possible solutions. Exploring alternative approaches remains an avenue for future research beyond the scope of this work. These two limitations highlight opportunities for future investigation.

