# OpenReview forum: "Navigating Concept Drift and Temporal Shift: Distribution Shift Generalized Time-Series Forecasting"
_ICLR.cc/2025/Conference — Submitted to ICLR 2025_

### Official Review · Reviewer_BM6V · 2024-10-17

**Soundness:** 3
**Presentation:** 2
**Contribution:** 2
**Rating:** 5
**Confidence:** 3

**Summary:**

This paper studies the problem of concept drifts and temporal shifts in the time-series forecasting tasks. To address the two problems, the algorithm, namely SHIFTS, is proposed. On one hand, SHIFTS puts forward the idea of SAM to address the concept drift problem. On the other hand, the problem of temporal shifts is combatted by the concept of ReIN,

**Strengths:**

1) The problems of concept drifts and temporal shifts are rarely considered in the time-series forecasting works.
2) the problem of time-series forecasting is important and has real-world relevance.

**Weaknesses:**

Literature Survey

literature survey of time-series forecasting is not rigorous. please include recent works listed below:

- FRNet: Frequency-based Rotation Network for Long-term Time Series Forecasting
- Fredformer: Frequency Debiased Transformer for Time Series Forecasting

Novelty

Novelty is rather limited because the temporal shift is addressed using ReIN. Can you clarify whether you have any modification to the vanilla ReIN method?

Experiments

1) please analyze whether the datasets themselves contain the concept drifts and the temporal shifts.
2) what is the time-series model that you used? what are the time-series models that the baseline algorithms used?
3) There are newer baselines for time-series forecasting. please compare your algorithm with them.

Conclusion

There is no limitation mentioned in the conclusion section.

**Questions:**

1) what are d_{x} and d_{y}? they are used before being defined.

2) what is the time-series model that you used? what are the time-series models that the baseline algorithms used?

3) can you confirm that the concept drift and the temporal shift are present in the dataset used in this study?

---

> ### Author Response · Authors · 2024-11-21
> **Rebuttal to Reviewer BM6V**
>
> **Response to Literature:**
>
> We believe we have covered most state-of-the-art literature in both time-series forecasting and distribution shift research, including recent works such as iTransformer (ICLR 2024) and FOIL (ICML 2024). Fedformer is already discussed in the current version, and we are willing to including FRNet in the discussion for completeness.
>
> **Response to Novelty:**
>
> First and most importantly, this work aims to show how to mitigate concept drift (SAM) and how to integrate concept drift method together with temporal shift (ShifTS). Finding better temporal shift methods is not the main scope of this work. We choose to use vanilla RevIN in ShifTS as it is simple yet effective. We admit there are advanced temporal shift methods, such as SAN or NS-Tansformer. However, these methods require either forecasting model modification or pre-training.
>
> Second, we want to clarify that instance normalization is necessary for ShifTS. To mitigate concept drift where the conditional distribution $P(Y^H|X^L)$ is unstable, we need to first mitigate temporal shift and ensure the marginal distribution is stable, and the most effective way is by doing instance normalization, such that all marginal distributions ~P(0,1).
>
> We have made a modification to Section 4.3 for a more clear presentation.
>
> **Response to Experiment:**
>
> (**Also to Q3**) We showcased an example of concept drift and temporal shift in Figures 4, 5, and 6(a), in Appendix A. Also, the selected datasets are known for irregular patterns and have been used for previous distribution shift studies [1].
>
> (**Also to Q2**) The proposed ShifTS is model agnostic. Therefore, we compare two types of baselines: First, we show that with ShifTS, existing time-series forecasting models, such as Informer, can gain consistent improvements, as shown in Table 1. Second, we show ShifTS better handles distribution shift issues compared to distribution shift methods, including concept drift methods for general purposes (e.g., classification) such as IRM temporal shift methods for time series, such as RevIN, and combined methods, as shown in Table 2.
>
> For forecasting models, we are using up to iTransformer and TimeMixer of ICLR 2024. For distribution shift methods, we are using up to FOIL of ICML 2024. We believe the baselines we used are new enough, we would appreciate it if the reviewer could point out the wanted baselines.
>
> **Response to Conclusion:**
>
> Thanks for pointing this out. We have added a limitation discussion in Appendix D in the revised version.
>
> **Response to Q1:**
>
> $d_{x}$ and $d_{y}$ refer to the dimension of exogenous features and target features.
>
>
> [1] Time-Series Forecasting for Out-of-Distribution Generalization Using Invariant Learning. ICML 2024

---

> > ### Author Response · Authors · 2024-12-02
> > **Please Review the Response**
> >
> > Dear Reviewer BM6V,
> >
> > As the deadline for discussion is approaching, we encourage you to review our response and let us know if you have any further concerns.

---

### Official Review · Reviewer_2fAg · 2024-11-04

**Soundness:** 2
**Presentation:** 4
**Contribution:** 3
**Rating:** 6
**Confidence:** 4

**Summary:**

This paper introduces ShifTS, a framework that addresses both concept drift and temporal shift issues in time-series forecasting. (1) The work presents a novel soft attention mechanism (SAM) for concept drift and integrates it with temporal shift mitigation techniques. (2) demonstrates substantial empirical improvements across diverse settings, and (3) provides a practical, model-agnostic solution to an important problem in time-series forecasting.

**Strengths:**

1. The SAM mechanism effectively leverages horizon information to mitigate concept drift
The framework successfully combines concept drift and temporal shift solutions in a principled way
2. Comprehensive experiments across six datasets show consistent improvements
    1. Performance gains are substantial: up to 81.9% on ILI and 71.4% on ETTh2 (Table 1)
    2. Results hold across multiple state-of-the-art models, demonstrating generalizability
    3. Ablation studies clearly demonstrate the value of addressing both types of shifts
3. Model-agnostic design enables broad applicability (Implementation is straightforward with minimal to no hyperparameter tuning required)
4. The comparison against other distribution shift methods (Table 2) demonstrates superior performance
5. The theoretical framework linking concept drift to insufficient determination by X_L is well-motivated
6. The paper addresses a significant gap in time-series forecasting by tackling both concept drift and temporal shift

**Weaknesses:**

1. Section 4.3 on "MITIGATING TEMPORAL SHIFT" lacks novelty:
    1. Simply restates RevIN equations (4 and 5) without adding new insights
    2. It should be significantly condensed or moved to background/preliminaries, as currently, it merely applies the existing RevIN method.
    3. It takes space that could be better used to explain novel contributions
2. Practical feasibility concerns with exogenous feature forecasting:
    1. The paper proposes forecasting X̂_SUR to support target series prediction
    2. This assumes we can accurately forecast exogenous features, which may be as hard as the original problem. No clear justification for why forecasting exogenous features would be more reliable than direct target forecasting. The surrogate loss (equation 3) doesn't guarantee accurate exogenous forecasts.
    3. The method likely works on the benchmark datasets primarily because X itself has limited concept drift in chosen long-term benchmarking datasets (Limited dataset diversity (ILI, Exchange, ETT share similar characteristics). Success may not generalize to scenarios with significant concept drift in exogenous features themselves.
    4. Consider synthetic experiments with controlled concept drift in X.
3. The performance gains of ShifTS over baseline methods lack convincing statistical validation:
    1. Performance differences between ShifTS and baseline methods in Table 2 are inconsistent and often marginal (e.g., on ETTh1: ShifTS MSE 0.076 vs FOIL 0.081 vs RevIN 0.085, and on ETTh2: ShifTS 0.194 vs SAN 0.199 vs N-S Trans 0.203), with no statistical significance testing or variance reporting across seeds to validate these small improvements.
    2. ShifTS's gains over strong baselines like SAN and FOIL appear dataset-dependent, and the margins are small enough to question statistical significance—especially noticeable on Exchange, where temporal shift methods perform comparably (ShifTS 0.470 vs SAN 0.415) and ShifTS+SAN (0.407) perform better than ShifTS alone.
    3. Without proper variance reporting across different model seeds, it's difficult to conclude if ShifTS truly outperforms existing methods or if the improvements (often 2-5% range) fall within standard deviation ranges of multiple training runs
4. The choice of sparsity regularization in equation (2) needs better motivation
5. The mutual information analysis (Figure 3a) could benefit from more rigorous statistical validation beyond the p-value.

Minor comments
1. While the approach is model-agnostic, all evaluations use transformer-based models. Testing with other architectures (e.g., RNNs) would strengthen the claims.

**Questions:**

1. Statistical Validation:
    1. Could you provide variance analysis across different random seeds and training runs, especially for the marginal improvements (2-5%) shown in Table 2?
    2. How do you justify the statistical significance of improvements over baselines like SAN and FOIL given the small margins?
2. How does ShifTS handle scenarios where exogenous features themselves exhibit strong concept drift? Can you provide empirical evidence showing why forecasting X̂_SUR is more reliable than direct target forecasting? How do poor exogenous feature forecasts impact the final prediction quality?
3. Why were only transformer-based models used despite claims of model-agnosticism?
4. Could you explain how the chosen datasets (ILI, Exchange, ETT) represent different types of concept drift?
5. Why does ShifTS+SAN outperform ShifTS alone on the Exchange dataset? Does this suggest limitations in your approach?

---

> ### Author Response · Authors · 2024-11-21
> **Rebuttal to Reviewer 2fAg**
>
> **Response to W1:**
>
> Thanks for pointing this out.
>
> First and most importantly, this work aims to show how to mitigate concept drift (SAM) and how to integrate concept drift method together with temporal shift (ShifTS). Finding better temporal shift methods is not the main scope of this work. We choose to use vanilla RevIN in ShifTS as it is simple yet effective. We admit there are advanced temporal shift methods, such as SAN or NS-Tansformer. However, these methods require either forecasting model modification or pre-training.
>
> Second, we want to clarify that instance normalization is necessary for ShifTS. To mitigate concept drift where the conditional distribution $P(Y^H|X^L)$ is unstable, we need to first mitigate temporal shift and ensure the marginal distribution is stable, and the most effective way is by doing instance normalization, such that all marginal distributions ~P(0,1).
>
> We agree with the reviewer’s comment on Section 4.3. We have modified Section 4.3 with the aforementioned intuitions for a clearer presentation.
>
> **Response to W2:**
>
> First, while we agree that forecasting the whole horizon window of the exogenous features can be as hard as the original problem, this work does not aim to forecast $X^H$ using $X^L$. Instead, it uses $X^L$ to forecast $X^{SUR}$, which is a simpler task as follows:  First, forecasting $X^{SUR}$ does not need to estimate the values of the whole horizon window, but only some key points showing invariant patterns (as shown by the purple circles in the add-on case study at:https://anonymous.4open.science/r/shifts_iclr-56A0/case_study.pdf), this is generally easier; Second, invariant patterns can exist in both look back and horizon windows, with similar representations, finding these invariant patterns and using representations in the loopback window to forecast similar representations in the horizon is easier. This misunderstanding may come from an unclear presentation in Figure 1, where we have uploaded a clearer version to avoid this misinformation.
>
> Second (**Also to Q4**), we want to clarify two points in the reviewer’s comment: (1) Concept Drift vs. Temporal Shift: It seems there may be some confusion between the definitions of concept drift and temporal shift. In this work, concept drift specifically refers to changes in the correlation between exogenous features and targets. Therefore, we only consider temporal shifts within a single exogenous feature. (2) Temporal Shift in the Dataset: The dataset used in this study does exhibit temporal shift, as illustrated in Figures 4 and 6(a). Datasets such as ETT are known for their irregular patterns, which effectively demonstrate the effectiveness of our methods in handling distribution shifts. Furthermore, the use of these datasets is consistent with prior studies on distribution shift [1].
>
> Third, we are willing to add synthetic experiments with controlled concept drift and temporal shift. The results still show the benefits of ShifTS. Detailed setup and results are shown at: https://anonymous.4open.science/r/shifts_iclr-56A0/synthetic_exp.pdf
>
> **Response to W3 & Q1:**
>
> We run the experiments on three different random seeds and take the average, following a typical experiment and report manner as common in previous studies. For Table 2, we additionally average the forecasting results across four horizon windows. We believe the results are statistically meaningful.
>
> Additionally, we are able to show the results of Table 2 before averaging at: https://anonymous.4open.science/r/shifts_iclr-56A0/expand_table2.pdf. The results and benefits of ShifTS are consistent with the averaged results in Table 2.
>
> **Response to W4:**
>
> The motivation for sparsity regularization corresponds to lines 202~205. We believe higher weights refer to the invariant patterns, while the lower weights refer to the variant patterns. We use sparsity regularization to filter the variant patterns whose correlations to target are more likely to change over time.
>
> **Response to W5:**
>
> Figure 3 aims to indicate when to use ShifTS, and when ShifTS is more beneficial. We believe the p-value test is strong enough to show a positive correlation between MI and performance gains. Maybe the reviewer can point out other statistical tests that are wanted.

---

> > ### Author Response · Authors · 2024-11-21
> > **Rebuttal to Reviewer 2fAg (Cont.)**
> >
> > **Response to Minor Comment & Q3:**
> >
> > While Transformer-based models are commonly used for these datasets, we are willing to include RNN results in the following table on ETTh1/ETTh2/Exchange 96-96, and ILI 36-24 (Note that RNN’s sequential natural does not allow horizon window longer than lookback). Our method still shows benefits with RNN.
> >
> > |          | RNN   |       | RNN+ShifTS |           |
> > |----------|-------|-------|------------|-----------|
> > |          | MSE   | MAE   | MSE        | MAE       |
> > | ETTh1    | 0.658 | 0.756 | **0.065**  | **0.196** |
> > | ETTh2    | 1.246 | 1.031 | **0.167**  | **0.322** |
> > | ETTm1    | 0.089 | 0.229 | **0.029**  | **0.129** |
> > | ETTm2    | 0.220 | 0.380 | **0.084**  | **0.221** |
> > | Exchange | 1.224 | 0.926 | **0.119**  | **0.263** |
> > | ILI      | 3.017 | 1.494 | **0.768**  | **0.663** |
> >
> > **Response to Q2:**
> >
> > In addition to the intuitions provided in our response to W2, we have demonstrated the benefits of forecasting with $\hat{X}^{SUR}$ in the main results Tables 1 and 2, which show reduced forecasting errors.
> >
> > Furthermore, we are willing to illustrate that the selected surrogate features $X^{SUR}$ are ‘good’ features for forecasting the target. We compare the performance difference between forecasting $Y^H$ with $\hat{X}^{SUR}$ (practical scenario), and $X^{SUR}$ (ideal scenario, use to show benefits of $X^{SUR}$). The results on the ETT datasets, available at:https://anonymous.4open.science/r/shifts_iclr-56A0/effective_mask.pdf, clearly demonstrate the benefits of forecasting with $X^{SUR}$.
> >
> > **Response to Q5:**
> >
> > No, our work's main contributions lie in (1) Addressing concept drift in time series (SAM) and (2) Integrating concept drift and temporal shift methods as a unified framework (ShifTS). Developing advanced temporal shift methods, such as SAN, is not the scope of this work. Our goal is to show if SAN properly addresses temporal shift, using SAM and ShifTS to additionally address concept drift can make forecasting performance even better.

---

> > > ### Comment · Reviewer_2fAg · 2024-11-27
> > >
> > > I thank the authors for their thorough responses and additional experiments addressing my concerns. While these clarifications strengthened the paper, I maintained my score of 6 as I still have concerns.
> > >
> > > Outstanding concerns:
> > >
> > > 1. Statistical validation remains inadequate: Using only three random seeds is insufficient for robust validation
> > > The expanded Table 2 merely shows averaging across horizons rather than proper statistical analysis. No reporting of variance or confidence intervals across different training runs/seeds. This is especially important due to small performance margins in table 2 and over baselines like SAN and FOIL (2-5% range) that need rigorous statistical validation to confirm significance.
> > >
> > > Positive developments:
> > >
> > > 1. Explains exogenous feature invariant X-SUR's forecasting mechanism through case studies.
> > > 2. New RNN experiments demonstrating model-agnostic properties
> > > 3. New synthetic experiments look really good, supporting their hypothesis.

---

> > > > ### Author Response · Authors · 2024-12-02
> > > > **Response to Additional Comments**
> > > >
> > > > We thank the reviewer for the positive feedback!
> > > >
> > > > To further address the reviewer's concern, we are willing to show the results on SAN, FOIL,  and ShifTS with std reported. The detailed results are shown at: https://anonymous.4open.science/r/shifts_iclr-56A0/table2_stats.pdf. We believe the improvement of ShifTS is notable with the t-test validated on each evaluation.

---

### Official Review · Reviewer_vEmg · 2024-11-05

**Soundness:** 2
**Presentation:** 3
**Contribution:** 2
**Rating:** 3
**Confidence:** 5

**Summary:**

This paper presents a generalized solution for (univariate) time-series forecasting with considerations of both concept drift and temporal shift. The idea of introducing concept drift to time series forecasting is interesting. The authors have clearly presented the main idea of the proposed framework, aka, ShifTS, with comprehensive experiments of adapting this framework to recent time series methodologies.

**Strengths:**

The idea of proposing a generalized framework for time series forecasting is both interesting and inspiring.

The framework itself is well-conceived and is supported by robust experimental validation.

Additionally, the writing is clear and well-organized, making the work accessible and easy to follow.

**Weaknesses:**

Although it is interesting to propose a generalised framework that can handle both temporal shift and concept drift, the proposed solution of how they are combined lacks novelty, neither with in-depth insights into these different distributional changes. That is to say, I can understand why this framework works very well, but from my perspective, its potential to inspire further research is somewhat limited. This, combined with the fact that this is not the first study to introduce concept drift in the time series forecasting domain, largely informs my overall evaluation of this submission.

**Questions:**

Referring to your definitions:

Definition 3.1 (Temporal Shift (Shimodaira, 2000; Du et al., 2021))
Temporal shift (also known as virtual shift (Tsymbal, 2004)) refers to changes in the marginal probability distributions over time, while the conditional distributions remain the same.

Definition 3.2 (Concept Drift (Gama et al., 2014; Lu et al., 2018))
Concept drift refers to changes in the conditional distributions over time, while the marginal probability distributions remain the same.

Let's take the definition of concept drift from Gama et al., 2014 here:
In their equation (2), "Formally, concept drift between time point t0 and time point t1 can be defined as ∃X: pt0 (X, y)= pt1 (X, y)" which is defined on the joined distribution rather than just the changes in the conditional distributions over time.

This means that if you’re using this definition, it inherently covers changes in marginal probability distributions over time—essentially encompassing what you defined as temporal shift. This overlap is also acknowledged in your mention of “Temporal shift (also known as virtual shift (Tsymbal, 2004)),” with Tsymbal’s work (2004) being a notable study in the concept drift domain.

Therefore, the problem setting and definitions require major revision to ensure rigor, which does not meet the ICLR standards for acceptance.

---

> ### Author Response · Authors · 2024-11-21
> **Rebuttal to Reviewer vEmg**
>
> **Response to Weakness:**
>
> We respectfully disagree with the comments from the reviewer regarding the novelty.
>
> First, as the reviewer him/herself mentioned in the weakness, the proposed framework is interesting and is understandable for why it works well, we believe this work’s idea is intuitive, simple, and effective.
>
> Second, while general concept drift problems are widely studied, studying concept drift issues in time-series forecasting is largely overlooked with existing methods showing certain limitations, as mentioned in our paper lines 54~59.
>
> **Response to Question:**
>
> Thanks for pointing out the definition questions. We wanted to clarify - there is just some confusion in the terminology here. Gama et al paper does use Equation 2 as a definition of ‘concept drift’. However, this is too general.
>
> Indeed in the same paper (Gama et al) immediately after Equation [2], they elaborate that there are two types of concept drift:
>
> (1) “Real concept drift refers to changes in p(y|X)”, which we refer to simply as ‘concept drift’ in our work - this terminology is also used in e.g. in [2]; and
>
> (2)  “Virtual drift happens if the distribution of the incoming data changes without affecting p(y|X)” (change in p(X)), which we refer to as ‘temporal shift’ in our work, and this terminology is also used in [3].
>
> Hence this is just a matter of using slightly different terminology than Gama et al. - we have arguably used terms in our paper that are more popular in the time-series domain [2, 3]. Therefore, we respectfully disagree with the reviewer’s comment that the definitions in our work are not rigorous. Also, to meet the reviewer's expectations, we have used a more rigorous definition in the updated version.
>
> [1] A survey on concept drift adaptation
>
> [2] FEDD: Feature Extraction for Explicit Concept Drift Detection in Time Series
>
> [3] AdaRNN: Adaptive Learning and Forecasting of Time Series

---

> > ### Comment · Reviewer_vEmg · 2024-11-26
> >
> > Thanks for your answer.
> >
> > In your new target $P(Y^H ) = P(Y^H |Y^L)P(Y^L) + P(Y^H |X^L)P(X^L)$, the right:
> >
> > $P(Y^H|Y^L)P(Y^L)$ should be $P(Y^H,Y^L)$, and
> >
> > $P(Y^H|X^L)P(X^L)$ should be $P(Y^H,X^L)$.
> >
> > Could you give me the conditions that support $P(Y^H ) = P(Y^H,Y^L)+P(Y^H,X^L)$ ?
> >
> > Here H and L can be considered as different collects of values of t, right? In your real drift, $P(Y^H_i|X^L_i)\neq P(Y^H_j|X^L_j)$, somtimes your $i,j,H, L$ will have overlaps, right?
> >
> > This then takes a role when you are designing your methodology. However, I didn't see such details. That's why I am not clear.
> >
> > For me, it is important to understand **why** your method is working well during the experiments. Because of this lack of details, in my understanding, why good experiential performance can still be explained within the existing studies. So, I don't think this work is novel.
> >
> > You claim forecasting can be addressed by considering the cross-effect between concept drift and temporal shift while existing ones have discussed considering concept drift and considering temporal shift only. To me, the first work that introduced concept drift to time series forecasting is a work that addresses concept drift and temporal shift together because as you mentioned temporal shift has been broadly discussed. You didn't provide a novel methodology to support your experiment merit, compared to existing findings.
> >
> > It is interesting that revising the definition that you used in this methodology does not impair the methodology. Does this mean, this method is less rigorous when it is originally proposed?

---

> > > ### Author Response · Authors · 2024-12-02
> > > **Response to Additional Comments**
> > >
> > > We thank the reviewer for carefully reviewing our response!
> > >
> > > **Response to: Could you give me the conditions that support $P(Y^H ) = P(Y^H,Y^L)+P(Y^H,X^L)$**
> > >
> > > Our Figure 1 (Left) describes the causal relationships between $X^L, Y^L, Y^H$, which follows a natural belief that $Y^H$ is not only caused by its autocorrelations of $Y^L$ but also the correlations of exogenous features $X^L$. This formula simply describes this relationship, following the total probability theorem.
> > >
> > > **Response to method intuition**
> > >
> > > First, yes, there could be overlapping between samples, for example, if $j=i+1$, then there are $L-1$ overlaps between $X^L_i$ and $X^L_j$.
> > >
> > > Second, we provide intuitions of how our real concept drift method operates in lines 196–206 of Section 4.2 and why the temporal shift method is necessary in the revised Section 4.3, which together form the intuitions behind ShifTS. These explanations are also provided in our responses to other reviewers (e.g., N1TG).
> > >
> > > To clarify further, the intuition behind ShifTS can be summarized as follows: Given our assumption that both marginal and conditional distributions may be unstable in time-series forecasting, we begin by applying normalization and denormalization to ensure the marginal distributions align with a standard 0-1 distribution. For instance, after normalization, distributions such as $P(Y^H)$ and $P(Y^L)$ or $P(X^L)$ are standardized. This normalization process, detailed in the updated Section 4.3, ensures that the marginal distributions become invariant. Subsequently, we employ the SAM framework to learn a stable conditional distribution. In SAM, we search and aggregate across both the lookback and horizon windows to identify invariant patterns over time, denoted as $X^{SUR}$. These patterns, which receive higher weights by SAM, are identified as stable invariant features. By focusing on $X^{SUR}$, the model mitigates the effects of real concept drift and maintains stable conditional distributions over time, thereby improving forecasting the target.
> > >
> > >
> > > **Response to novelty**
> > >
> > > First, we would like to clarify the timeline of studies addressing real concept drift and temporal shifts in the literature. While concept drift was introduced in time-series analysis prior to the 2020s, early studies focused on simple datasets and online settings, often without accounting for temporal shifts. Temporal shifts, in contrast, began to receive more attention after the introduction of the ETT dataset in the 2020s, with works such as RevIN (2021) and studies that are established on real-world datasets and standard time-series forecasting settings. However, these studies largely overlooked real concept drift. To the best of our knowledge, limited work remains that integrates real concept drift and temporal shifts within standard time-series forecasting frameworks using popular real-world datasets. This gap underscores the novelty of our work, as we are among the first to study these two challenges together in such settings. This is also mentioned in lines 53~59.
> > >
> > > Second, we want to highlight that the novel methodology of this work lies in SAM, which is our way of mitigating real concept drift, and ShifTS, which is our way of handling both temporal shift and real concept drift together. The only thing that is not novel is the temporal shift method, which has been broadly discussed. However, we still provide intuitions of why instance normalization is necessary in our revised version. Our empirical evaluations in Table 1,2 show the effectiveness of ShifTS, and Figure 3 (Right) shows the effectiveness of SAM through ablation studies. Therefore, we respectfully disagree with the statement, 'You didn't provide a novel methodology to support your experiment merit.'
> > >
> > > **Response to rigorousness**
> > >
> > > We respectfully disagree with the reviewer’s comment. We would like to clarify that we did not alter the fundamental points of the definition. Specifically, while our use of the term ‘concept drift’ is popular in the time-series literature, this terminology named 'real concept drift' by Gamma 2014 is suggested by the reviewer. We would like to clarify this to meet the reviewer’s expectations through slight modifications. However, this adjustment does not imply any error or modification in our underlying definition.

---

### Official Review · Reviewer_N1TG · 2024-11-08

**Soundness:** 3
**Presentation:** 4
**Contribution:** 3
**Rating:** 6
**Confidence:** 3

**Summary:**

This paper addresses the challenge of distribution shifts in time-series forecasting, specifically focusing on concept drift and temporal shift. The authors identify two types of distribution shifts: concept drift, where the conditional distributions change over time, and temporal shift, where the marginal distributions change. They note that while existing studies primarily focus on temporal shifts, concept drift in time-series data has received less attention. To mitigate concept drift, the paper introduces a novel soft attention mechanism called soft attention masking (SAM). SAM leverages exogenous information from the horizon window to weigh and ensemble time series at multiple horizon time steps, enhancing the model's generalization ability. For addressing both concept drift and temporal shift, the authors propose ShifTS, a model-agnostic framework that integrates SAM with established temporal shift mitigation techniques.

**Strengths:**

1. The proposed soft attention masking (SAM) looks interesting. It leverages exogenous information to mitigate concept drift by weighing and ensembling time series at multiple horizon time steps.
2. The proposed ShifTS also looks interesting to me. As a model-agnostic framework, it combines various methodologies effectively, resulting in a robust solution.
3. The proposed method is evaluated on 6 series datasets in comparison with several baselines.

**Weaknesses:**

1. In general, this paper presents how an algorithm works, but lacks in-depth explanation why it works.
2. I am very interested to see how the proposed algorithm works on noisy datasets.

**Questions:**

1. How does the performance of ShifTS vary across different types of time-series data, especially those with high noise levels or irregular patterns? I highly recommend to add some experiments on this.
2. Is there any plan to explore the application of ShifTS in real-time or online forecasting scenarios, where the model continuously updates with incoming data?

---

> ### Author Response · Authors · 2024-11-21
> **Rebuttal to Reviewer N1TG**
>
> **Response to W1:**
>
> We would like to provide the intuition of ShifTS as follows: Given our assumption and the fact that both marginal and conditional distributions can be unstable, in the context of time-series forecasting, we first do normalization and denormalization to ensure both marginal distributions follow 0-1 distribution, e.g. both $P(Y^H)$ and $P(Y^L)$ or $P(X^L)$ follow standard distribution after normalization. This intuition is given in the updated Section 4.3. With this process, we assume the marginal distribution becomes invariant and then learn the stable conditional distribution through SAM. In SAM, we search and aggregate over both the lookback and horizon window to find the invariant patterns over time ($X^{SUR}$). For the patterns in the horizon window ($X^H$), we use conventional time-series forecasting models to forecast them using $X^L$, and then use these invariant patterns to learn accurate targets.
>
> **Response to W2 and Q1:**
>
> First, we clarify that the datasets used in this study are inherently irregular and noisy, as illustrated in Figures 4, 5, and 6(a). These datasets have also been utilized in previous studies on distribution shifts [1].
>
> Second, to address the concern about the effectiveness of ShiftTS on noisy data, we are willing to include additional experiments conducted on synthetic datasets with additive noise. The results consistently demonstrate the advantages of ShiftTS. The detailed experimental setup and results are provided at: https://anonymous.4open.science/r/shifts_iclr-56A0/synthetic_exp.pdf
>
> **Response to Q2:**
>
> Thank you for pointing out the real-time forecasting setup. Sure, ShiftTS is applicable to online settings. To demonstrate this, we provide an initial evaluation of the ETTh1 and ETTh2 datasets, using a lookback window of 96, a horizon of 96, and retraining every 96 timesteps (with training data increasing accordingly). The results presented in the following table still show the benefits of ShiftTS in this context.
>
> | Dataset | MSE (w.o. ShifTS) | MAE (w.o. ShifTS) | MSE (w. ShifTS) | MAE (w. ShifTS) |
> |:-------:|:-----------------:|:-----------------:|:---------------:|:---------------:|
> | ETTh1   | 0.049             | 0.210             | **0.034**       | **0.184**       |
> | ETTh2   | 0.165             | 0.319             | **0.145**       | **0.315**       |

---

> > ### Comment · Reviewer_N1TG · 2024-12-02
> >
> > Thanks for the feedbacks. I acknowledge I have read the feedbacks and decide to maintain my score.

---

### Official Review · Reviewer_Sduk · 2024-11-08

**Soundness:** 3
**Presentation:** 2
**Contribution:** 2
**Rating:** 3
**Confidence:** 2

**Summary:**

The paper addresses the challenges of concept drift and temporal shift in time-series forecasting. The authors propose a model-agnostic framework called ShifTS, which integrates a soft attention mechanism (SAM) to handle concept drift and existing temporal shift mitigation techniques. The framework aims to enhance the generalization and forecasting accuracy of time-series models by addressing both types of distribution shifts simultaneously. Extensive experiments demonstrate that ShifTS outperforms existing methods across various datasets and models. However, the methods used to address concept drift lack sufficient persuasiveness, and the techniques for handling temporal shifts do not introduce significant innovation.

**Strengths:**

- The paper identifies and formulates the dual challenge of concept drift and temporal shift in time-series forecasting, which has been understudied in the literature.
- The use of extensive experiments across multiple datasets and models demonstrates the robustness and reliability of the proposed methods.
- The paper is well-structured .

**Weaknesses:**

- The formula $\mathrm{P}(\mathbf{Y}^H)=\mathrm{P}(\mathbf{Y}^H|\mathbf{Y}^L)\mathrm{P}(\mathbf{Y}^L)\mathrm{P}(\mathbf{Y}^H|\mathbf{X}^L)\mathrm{P}(\mathbf{X}^L)$appears in Section 3.2.  However, it is not clear why this equality holds. A detailed derivation or explanation of the probabilistic relationships involved would enhance the clarity and understanding of the formula.

- Figure 2 is very difficult to understand. It would be beneficial to provide a more detailed caption or a step-by-step explanation of the components and their interactions. Additionally, the term "Agg" appears in both Algorithm 1 and Figure 2 but is not defined or explained in the main text. Including a clear definition and explanation of "Agg" would improve the readability and coherence of the paper.

- The soft attention mask  $M$ is introduced, but it is not clear how the attention mechanism is implemented. Meanwhile, it is mentioned that $M$ is independent of $\mathbf{X}^L$ and $\mathbf{Y}^L$ during testing. This raises the question of how $M$ can effectively address concept drift in test data. Please explain how $M$, which is independent of the lookback data, can still mitigate concept drift in the test data.

- The paper aims to solve the problem of simultaneous temporal shift and concept drift in time-series forecasting. It is important to explain how the model handles the joint changes in both marginal and conditional distributions. Specifically, it should be clarified what distribution remains invariant and how the model learns to adapt to these changes. Given that handling both types of drifts is challenging even in traditional classification tasks, a detailed explanation of why and how the proposed method is effective in the time-series context would strengthen the paper.
- Besides performance metrics, additional evidence is needed to demonstrate how the soft attention mask
$M$  specifically mitigates concept drift.

**Questions:**

Refer to weakness.

---

> ### Author Response · Authors · 2024-11-21
> **Rebuttal to Reviewer Sduk**
>
> **Response to W1:**
>
> Thanks for pointing this out and sorry it is a typo. It should be $P(Y^H)=P(Y^H|Y^L)P(Y^L)+P(Y^H|X^L)P(X^L)$, then the equation follows a very fundamental statistical law (law of total probability, https://en.wikipedia.org/wiki/Law_of_total_probability). This formulation also follows Equation 1 in [1] and definitions in [2]. We correct this on the updated version.
>
> **Response to W2:**
>
> We would like to expand the caption of Figure 2 (Please refer to the updated version) and clarify the process as follows: First, ShifTS consists of normalization (a) in the beginning and denormalization (c) at the end to mitigate the temporal shift as RevIN [3] does. Second, for mitigating concept drift part, ShifT processes two-stage forecasting as (b): It first forecasts the surrogate exogenous features, determined by the SAM, which are stable and invariant patterns towards forecasting the target. It then uses both forecasted surrogate exogenous features and original $Y^L$ to predict $Y^H$.
>
> The module ‘Agg’ is a simple MLP that aggregates weighing the exogenous features to predict the target, as also explained in the updated version.
>
> **Response to W3:**
>
> First, $M$ is initialized uniformly. During training, $M$ is utilized to compute the softmax weights, and features with higher scores are selected as invariant patterns. $M$ is updated through standard backpropagation. The implementation code is provided.
>
> Second, to clarify, we do not state that $M$ is independent of $Y^L$ and $X^L$. Instead, we say ‘the mask high attention weights are recognized as invariant patterns, which remain unchanged during test time steps’, which means we use invariant correlations between $X$ and $Y$ to make accurate predictions during the test.
>
> **Response to W4:**
>
> Following the clarification of W2, given our assumption that both marginal and conditional distributions can be unstable in time-series forecasting, we first do normalization and denormalization to ensure both marginal distributions follow 0-1 distribution, e.g. both $P(Y^H)$ and $P(Y^L)$ or $P(X^L)$ follow standard distribution after normalization. With this process, we assume the marginal distribution becomes stable and then learn the invariant conditional distribution through SAM. We give detailed and deeper intuitions in the updated Section 4.3.
>
> **Response to W5:**
>
> First, we provide an intuition for how $M$ effectively mitigates concept drift. $M$ is trained to select features across lookback and horizon windows. Features assigned higher weights by $M$, when averaged over the training data, are likely to represent invariant patterns. These invariant patterns are used to construct the surrogate feature $X^{SUR}$, which is less likely to suffer from concept drift during testing.
>
> Second, we are willing to provide a case study to further demonstrate the features selected by $M$ tend to be invariant between training and testing. The case study is shown at: https://anonymous.4open.science/r/shifts_iclr-56A0/case_study.pdf
>
> [1] A Survey on Concept Drift Adaptation
>
> [2] Adarnn: Adaptive learning and forecasting of time series
>
> [3] Reversible Instance Normalization for Accurate Time-Series Forecasting against Distribution Shift

---

> > ### Author Response · Authors · 2024-12-02
> > **Please Review the Response**
> >
> > Dear Reviewer Sduk,
> >
> > As the deadline for discussion is approaching, we encourage you to review our response and let us know if you have any further concerns.

---

### Author Response · Authors · 2024-11-21
**Global Response on Updated Paper**

We sincerely thank all reviewers for their insightful comments, which have significantly improved the quality of our paper. We have uploaded a revised version, where the following modifications are highlighted in red:

1. Lines 87–88: Added additional literature as suggested by Reviewer BM6V.

2. Line 129: Corrected a typo as pointed out by Reviewer Sduk.

3. Definition 3.2: Reworded for greater rigor as suggested by Reviewer vEmg.

4. Figure 1: Improved presentation as suggested by Reviewer 2fAg.

5. Figure 2: Expanded the captions for better clarity, addressing Reviewer Sduk’s concerns.

6. Section 4.3: Enhanced the discussion to provide deeper intuitions, as suggested by Reviewers Sduk, 2fAg, and BM6V.

7. Lines 277–278: Clarified the explanation of "Agg," as suggested by Reviewer Sduk.

8. Appendix D: Added a discussion on the paper's limitations, as suggested by Reviewers vEmg and BM6V.

We would encourage the reviewers to go through these changes, and we are willing to address additional concerns if any.

---

### Meta-Review · Area_Chair_tZWp · 2024-12-23

**Metareview:**

Summary

The paper addresses two types of distribution shifts in time-series prediction: temporal shift, where data distributions evolve over time, and concept drift, referring to shifts in the target time-series distribution. While existing research focuses primarily on temporal shifts, this work emphasizes the challenges of concept drift and their simultaneous occurrence. To tackle these issues, the authors propose a model-agnostic solution that combines soft attention mechanisms, which adaptively focus on relevant time horizons, and normalization strategies to stabilize predictions. This integrated approach is designed to handle both types of shifts concurrently. Experiments demonstrate that the proposed method outperforms baseline models.

Strengths

The paper's strengths lie in its focus on a problem that remains underexplored in existing literature. The work is well-structured and supported by extensive experiments. The experimental results demonstrate significant performance gains over baseline models.

Weaknesses

The paper has several weaknesses that limit its impact. First, there are presentation clarity issues, though these may have been addressed in later revisions. The rationale behind why the proposed method works is not sufficiently explained, reducing its persuasiveness. Additionally, the method does not introduce significant innovation, as it largely builds on existing techniques without substantial advancements. There is also a lack of rigorous definitions for key concepts, which undermines the paper's foundational contributions and novelty.

Recommendation

While the paper tackles a relevant problem in time-series forecasting and demonstrates strong experimental performance, it falls short in several critical areas. The lack of rigorous definitions and clarity in presentation weakens the theoretical foundation. The proposed method lacks innovation. Furthermore, the paper fails to persuasively explain why the proposed method works. These issues including the lack of foundational rigor and originality, suggest the paper is not ready for publication in its current form.

**Additional Comments On Reviewer Discussion:**

During the discussion phase, the authors made notable progress in addressing the presentation and clarity issues raised in the initial review. The revised paper and rebuttal clarified several aspects. However, the key concern regarding the lack of a thorough explanation of why the proposed method works remains unresolved. While the authors described how the algorithm operates and some of its assumptions (e.g., normalization ensuring marginal distribution invariance and higher weights identifying stable invariant features), these explanations remain heuristic and lack rigorous justification. The arguments provided are somewhat arbitrary and do not fully establish the soundness of the proposed approach, leaving concerns unsolved about the depth of the theoretical contribution. As a result, the reviewers' ratings and overall evaluation remain unchanged.

---

### Decision · Program_Chairs · 2025-01-22

Reject